# Exchanging registered users' submitting reviews towards trajectory privacy preservation for review services in Location-Based Social Networks

**Yunfeng Wang**[1,3]*, **Mingzhen Li**[1,2], **Yang Xin**[1,3]*, **Guangcan Yang**[1], **Qifeng Tang**[4], **Hongliang Zhu**[1], **Yixian Yang**[1,3], **Yuling Chen**[3]*

**1** School of Cyberspace Security, Information Security Center, National Engineering Laboratory for Disaster Backup and Recovery, Beijing University of Posts and Telecommunications, Beijing, China, **2** School of Computer and Information Engineering, Hechi University, Guangxi, China, **3** State Key Laboratory of Public Big Data, College of Computer Science and Technology, Guizhou University, Guiyang, China, **4** National Engineering Laboratory for Big Data Distribution and Exchange Technologies, Shanghai Data Exchange Corporation, Shanghai, China

* ylchen3@gzu.edu.cn (YC); yangxin@bupt.edu.cn (YX); wangyunfeng@bupt.edu.cn (YW)

**Data Availability Statement:** All relevant data are within the paper and its Supporting information files.

**Funding:** This research was funded by the "Major Scientific and Technological Special Project of

## Abstract

In Location-Based Social Networks (LBSNs), registered users submit their reviews for visited point-of-interests (POIs) to the system providers (SPs). The SPs anonymously publish submitted reviews to build reputations for POIs. Unfortunately, the user profile and trajectory contained in reviews can be easily obtained by adversaries who SPs has compromised with. Even worse, existing techniques, such as cryptography and generalization, etc., are infeasible due to the necessity of public publication of reviews and the facticity of reviews. Inspired by pseudonym techniques, we propose an approach to exchanging reviews before users submit reviews to SPs. In our approach, we introduce two attacks, namely review-based location correlation attack (RLCA) and semantic-based long-term statistical attack (SLSA). RLCA can be exploited to link the real user by reconstructing the trajectory, and SLSA can be launched to establish a connection between locations and users through the difference of semantic frequency. To resist RLCA, we design a method named User Selection to Resist RLCA (USR-RLCA) to exchange reviews. We propose a metric to measure the correlation between a user and a trajectory. Based on the metric, USR-RLCA can select reviews resisting RLCA to exchange by suppressing the number of locations on each reconstructed trajectory below the correlation. However, USR-RLCA fails to resist SLSA because of ignoring the essential semantics. Hence, we design an enhanced USR-RLCA named User Selection to Resist SLSA (USR-SLSA). We first propose a metric to measure the indistinguishability of locations concerning the difference of semantic frequency in a long term. Then, USR-SLSA can select reviews resisting SLSA to exchange by allowing two reviews whose indistinguishability is below the probability difference after the exchange to be exchanged. Evaluation results verify the effectiveness of our approach in terms of privacy and utility.

Guizhou Province (20183001)", the "Foundation of Guizhou Provincial Key Laboratory of Public Big Data (2017BDKFJJ015, 2018BDKFJJ008, 2018BDKFJJ020, 2018BDKFJJ021)", and the "Basic Ability Improvement Program for Young and Middle-aged Teachers of Guangxi (2021KY0615 and 2021KY0620).

**Competing interests:** No.

## Introduction

Recently, Location-Based Social Networks (LBSNs) [1] have become the dominant way people share information with others in our daily life, due to the rapid development of online social networks and the Location-Based Service. As an important component of an LBSN, Local Business Service Systems(LBSSs), such as Yelp, Tripadvisor, Dianping, etc., provide a review service [2]. In these systems, a registered user publishes a review each time she visits a point-of-interest (POI) and enjoys the service provided by the POI. Here, a POI is a business or shop registered in an LBSS. Note that the term 'user' in this paper refers to the registered user. Users hope to build reputations for POIs by publishing reviews. For example, a user altruistically publishes a good review since she has enjoyed a good service of a restaurant so that more people can enjoy the service. In general, the review has the following features:

- A POI and its location (geographic coordinates) correspond one-to-one. As we know, in an LBSS, although a POI is an area containing many geographic coordinates, the LBSS only selects a constant one as the location of the POI. Therefore, a review for a POI is equivalent to a review for the location.

- Facticity. Users are altruistic. They hope to build objective reputations for POIs by publishing reviews and do not mind who published reviews. Thus, all reviews are not fabricated and are published by the registered users who have enjoyed the service provided by a POI.

- Real time. Generally, most users publish reviews in a short time after visiting POIs and enjoying services. That is, visiting and publishing reviews for them are at the same time.

- Historicity. The POIs reviewed by a user during a period of time form a time-dependent trajectory. Generally, a user repeatedly and sequentially reviews some POIs, since she has a consistent lifestyle [3]. For example, she goes to a fixed restaurant for lunch at 11:00 am and a fixed cinema at 17:00 every day.

After enjoying the service of a POI, a user publishes her review through a two-step process. In the first step, a user submits her reviews to the LBSS server. In the second step, the LBSS server anonymizes and publishes the reviews while storing the reviews. We name the two steps as submit reviews (SR) and publish reviews (PR), separately. For example, Alice's name is anonymous as A*. The typical LBSS architecture is shown as in Fig 1. In the LBSS server, the user registration information, such as cell phone number and ID card, and reviews content are unencrypted and not anonymous. That is, for anonymously published reviews, users' identities are not anonymous to the SPs. After compromised with the SPs, the adversary can obtain user registration information and reviews, and then easily correlate the user identity.

Currently, privacy leakage, especially the trajectory privacy, has become one of the important challenges users face when using review services in LBSSs. For the one hand, because of

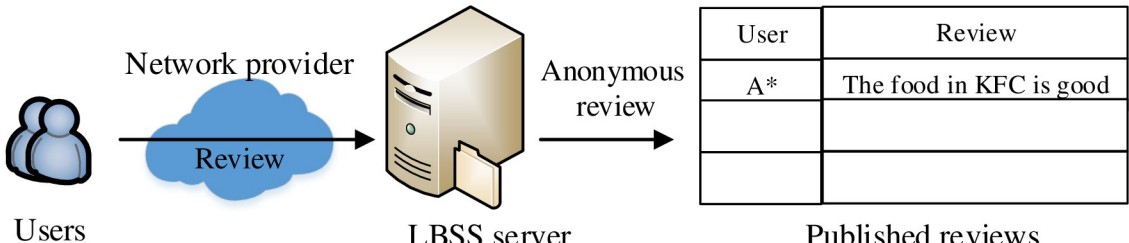

**Fig 1. The typical LBSS architecture.**

the one-to-one correspondence between the POI and the location, locations contained in reviews are inevitably disclosed when users publish reviews for POIs. To build objective reputations for POIs, a user must disclose the reviewed location in a review and ensure that review is not fabricated. For the other hand, a review for a POI indicates that the owner has visited this POI. The attacker can obtain the historical trajectory of the target user by collecting (such as cyber attack [4], information sharing [5], etc.) and analyzing the historical review data of the target user. More seriously, some locations of the historical trajectory may be sensitive for the owner, e.g., home, workplace, hospital, etc. Although existing works [2, 6] are capable of location privacy, they mainly focus on the PR scenario, not the SR. For example, by adopting a pseudonym, literature [2] makes it impossible for the adversary to know who publishes the reviews. However, in the SR scenario, the adversary who the system providers (SPs, i.e., the founders of Dianping or Yelp) has compromised with can obtain users' historical data that mainly include users' real identities and trajectories. The adversary can also analyze a user's trajectories and obtain her mobility pattern (e.g., Alice likes to go shopping after eating McDonald's at every Tuesday evening), even predict her mobility [7]. Addition, LBSSs requires enough reviews to innovate the system and improve services, but the more reviews are published, the more users' privacy is compromised. To protect trajectory privacy, an effective method is to build the non-correspondence between a user and her trajectories.

In our scenario, using existing technologies, e.g., cryptography, generalization, suppression, etc., for trajectory privacy protection is a challenge. Cryptography-based techniques are infeasible since users' trajectories are visible for adversaries in the LBSS. The $k$-anonymity hides the location among $k-1$ fabricated locations or a cloaking area with $k-1$ historical locations. While all locations in the LBSS are true and contain the spatial-temporal correlation. Suppression technique, which enables reviews that may reveal the user's privacy being invisible, is technically feasible since it does not have the disadvantages of the above two technologies. However, too many reviews being suppressed will reduce the utility, i.e., suppressing reviews may result in objective reviews being suppressed and non-objective reviews being published, which will build a reputation that cannot reflect the real service quality of the business. Considering the shortcomings above, one feasible solution is to assign each review a pseudonym. However, we can only assign a pseudonym chosen from the names set of users to a review since the SPs can obtain all users' identities information. In this paper, we propose a mechanism to exchange reviews of different users. Note that when talking about exchanging reviews between two users, we also call exchanging their locations that have been visited since a review corresponds to a POI(i.e., location). In this sense, each user is assigned a pseudonym. It can preserve users' trajectory privacy in the PR scenario and retain users' reviews, even though the SPs knows users' identity information and reviews.

Furthermore, since a human has a consistent lifestyle, a user's reviews always have spatial-temporal correlation and then the locations corresponding to the user's historical reviews form the user's movement trajectories. This situation is more vulnerable to link attacks. In this paper, we introduce two attacks, named review-based location correlation attack (RLCA) and semantic-based long-term statistical attack (SLSA). Firstly, as pointed out in [8, 9], adversaries can utilize the correlation to infer users' real identities, even though their identities are encrypted or anonymous. Moreover, we show that exchanging reviews can poorly protect users' real identities and reviews from being linked and the adversary can link the real user by reconstructing trajectories, which we call review-based location correlation attack (RLCA). Secondly, due to the consistent lifestyle, a human will periodically visit some locations that provide the same services (e.g., most people go to different restaurants near their workplace to eat at noon). It means that several locations with the same semantics will be visited more

frequently than others in a long term. The frequency difference of semantics can be utilized to establish a connection between locations and users. In this paper, we call this attack semantic-based long-term statistical attack (SLSA). As we show, SLSA can identify a user's real location among the exchanged locations with a high probability.

Basing on the above analysis, we present an approach to resist the above attacks. The approach contains two methods named User Selection to Resist RLCA (USR-RLCA) and User Selection to Resist SLSA (USR-SLSA). As we know, the adversary links a user and her trajectories by analyzing correlation. We use USR-RLCA to resist RLCA. In USR-RLCA, we propose a metric to measure the correlation between them. Based on the metric, we suppress the number of locations on each reconstructed trajectory below a threshold. Compared with existing methods, USR-RLCA can significantly prevent a user and her trajectories from being linked. Yet, due to ignoring the essential semantics, USR-RLCA fails to protect locations from being indistinguishable against SLSA. Hence, we propose USR-SLSA to solve this problem. In USR-SLSA, we first propose a metric that measures how indistinguishable different locations are concerning the frequency difference of semantics in a long term. Then, we select some reviews as a group for each review that will send to the SPs. In the group, users exchange their reviews based on the above metric. Two reviews are allowed to exchange if the probability difference of their semantics is below a threshold after the exchange. It ensures that this method can resist SLSA. We conduct experiments to evaluate the effectiveness of our approach in terms of privacy and utility. Results show that our approach can preserve users' privacy against RLCA and SLSA and outperform existing methods.

Besides user privacy, another issue that needs to be considered is whether our approach excessively reduces user reviews that can be published publicly, i.e., the user utility. The existing technique [2] to protect user privacy mainly limits the reviews that are publicly released. Different from it, our approach submits only reviews that do not reveal the trajectory privacy to the SPs, and does not focus on how the SPs publish the reviews. To evaluate the user utility, we use $(\epsilon,\delta)$-public principle [2] to publish reviews and use the ratio of the public reviews to measure it. We show that even though our approach submits fewer reviews to the SPs than [2], it does not reduce the user utility. The reason is that $(\epsilon,\delta)$-public principle would allow a higher ratio of reviews to be published if the SPs receive fewer reviews.

In summary, the major contributions of this paper are as follows:

- We propose a mechanism to preserve users' trajectory privacy in the PR scenario and retain user reviews. To the best of our knowledge, this is the first paper to investigate how to protect the privacy in the scenario, which users' identities and reviews are unencrypted and not anonymous to the adversary.

- According to the consistent lifestyle of a human, we introduce SLSA, which can exploit the frequency difference of semantics to establish a connection between locations and users. We also propose two methods to resist RLCA and SLSA.

- We propose two metrics that measure the correlation between a user and a trajectory and the indistinguishability of locations concerning the difference of semantic frequency in a long term, separately. Using them, we design USR-RLCA to suppress the number of locations on each reconstructed trajectory below a threshold, and USR-SLSA to ensure that the probability difference of semantics is below a threshold after users' reviews exchanging.

- The effectiveness of our methods in terms of the privacy and utility is verified on a real dataset. Results show that our methods can preserve users' privacy against RLCA and SLSA and outperform existing methods in terms of the utility.

## Related work

With growing concerns for privacy arising from prevalent LBSNs, many approaches have been proposed to protect user trajectory privacy. According to the way of protecting trajectory privacy, these approaches can be divided into four categories: cryptography, generalization, suppression and pseudonyms.

Cryptography mainly encrypt and make user privacy information invisible to the adversary [10, 11]. However, in the LBSSs, users' trajectories are visible for the adversary, since the reviews are public. To overcome the vulnerability, cryptography can encrypt users' other privacy information. In [12], a user's pseudo-ID is encrypted by using a symmetric encryption algorithm. In [7], the communication between the user and the LBS server is encrypted. Nevertheless, the adversary can still infer users' real identities by analyzing the spatial-temporal correlation between locations in trajectories [9, 13].

Generalization protects user trajectory privacy by hiding the user's actual trajectory (identity) among other users' trajectories (identities). The $k$-anonymity is one of the most widely used methods of generalization, which includes two primary approaches of Dummy Trajectory and Historical Trajectory. In Dummy Trajectory, for each location, methods [14–18] fabricated $k − 1$ locations to send to the LBS server. For example, method [17] randomly selects $k − 1$ dummy locations near the real location; method [18] rotates the real trajectory by an angle as a fake trajectory. In fact, the fabricated locations in [18] may not have been visited by anyone. Hence, using Dummy Trajectory, we can still review on unvisited locations. This violates the facticity of the reviews. In Historical Trajectory [19–22], for each location, methods do not fabricate, but sample $k − 1$ visited locations from historical data. For example, methods [19, 20] sample $k − 1$ complete trajectories to achieve k-anonymity; method [21] samples $k − 1$ trajectories and extends $k$ locations sampled from $k$ different trajectories into the same cloaking area; method [22] samples segments of trajectories to combine $k − 1$ trajectories. But, like the cryptography, Historical Trajectory is not workable to solve our problem, since sampling trajectories contain the spatial-temporal correlation of locations.

In the suppression, methods [19, 21, 23, 24] can make users' trajectories undistinguishable from the adversary by suppressing each user's personalized locations different from others. These methods are used to limit the published trajectory data. For example, the method [24] suppresses the sensitive or often visited locations in the trajectories; method [21] extends $k$ personalized locations in $k$ different trajectories into a cloaking area. A few works have studied users' privacy under conditions of non-personalized locations. The authors [19] limit the locations that are used to reconstruct the user trajectories to send to the LBS server. In particular, paper [2] set the threshold that the number of the user public reviews, so that the user reviews for a POI cannot all be published. However, suppressing too many locations or reviews will reduces the user utility.

The existing works [7, 9, 12] are based on pseudonyms. In [7], a user stores several user names and selects one of them as the current user name requesting LBS service at each query. In [9], each user has a pseudo name. If necessary, after consultation, all users replace their old pseudonyms with new pseudonyms while simultaneously restarting to use the LBS. Method [12] protects user identities from the attacker's recognition through exchanging their identities. Although pseudonym is infeasible for our problem due to the facticity of review, inspired by [12], we exchange reviews of different users to break the spatial-temporal correlation of locations.

There has been much work [25–30] on inferring attacks and metrics for privacy preservation. In [7], the authors proposed the long-term statistical attack (LSA). In [3], the authors point out the semantic similarity of locations. Based on [3, 7], we demonstrate the semantic-

based long-term statistical attack (SLSA). Besides, methods [8, 25, 26] and methods [7, 15, 27] measure the ability of preserving privacy using *k*-anonymity metric and entropy, respectively.

## Preliminary

### System model and basic boncepts

Assume an LBSS has *M* POIs denoted as {$POI_1$, $POI_2$, $\cdots$, $POI_M$} and *N* users denoted as {$u_1$, $u_2$, $\cdots$, $u_N$} in a city area. Though in theory each POI $POI_i$ has a one-to-one mapping with a unique geographic coordinate, one geographic coordinate can locate multiple POIs because of the low precision. For example, (lat:39.959679, lon:116.362065) is the FengLan International Shopping Center in Beijing, while (lat: 39.958624, lon: 116.363542) and (lat: 39.958744, lon: 116.363428) represent Watsons and Mothercare in the Shopping Center respectively. In many circumstances, some businesses or shops have the same name, such as chain shops. So, we use a unique five-tuple to define the POI:

Definition 1: A POI is a five-tuple as *POI* = < *stra*, *name*, *lon*, *lat*, *type* >. Here, *stra*, *name*, *lon*, *lat* and *type* represent structured address, name, geographic coordinate (often the longitude and latitude), and the semantic of the POI, respectively. The semantic refers to the type of service that a POI can provide, such as food, shopping, education, etc.

According to Definition 1, the POI in an LBSS can uniquely represent a business or shop with a geographical region in terms of physical location.

In the system, each user $u_j$ visits different POIs at different times. Each time, $u_j$ immediately reviews $POI_i$ after he visits it. In this paper, we use $POI_{ij}^{t_l}$ to denote $POI_i$ on which $u_j$ reviews at time $t_l$. Considering the spatial and temporal correlation among POIs, we give a formal definition of the trajectory as follows.

Definition 2: For a user $u_j$, her trajectory $T_j$ is a set of time-dependent discrete POIs reviewed in a cycle, which can be expressed as:

$$T_j = \{r_{ij} | i = 1, 2, \cdots, n\}$$

Where $r_{ij}$ is the *i*-th location on $T_j$ and denoted as a three-tuple $< POI(r_{ij}), t(r_{ij}), \tau(r_{ij}) >$, which means $u_j$ has visited and reviewed the $POI(r_{ij})$ at time $t(r_{ij})$ in period $\tau(r_{ij})$ in a cycle. In essence, a trajectory is a sequence of locations sorted in chronological order in which they are reviewed, e.g., $t(r_{1j}) \leq t(r_{2j}) \cdots t(r_{nj})$.

In an LBSS, each POI is uniquely represented by a 5-tuple. For each POI, users fill out and submit review to the SPs after logining account. The SPs can obtain each user's real identity and all reviews, due to the real-name registration. In reality, people always have a consistent daily life, means that a user always visits and reviews her most fixed places periodically, such as home, workplace, etc., different places where they engage in the same activity at the same period in different cycles(e.g., a user plays table tennis or badminton every night at 20:00). In this paper, we refer to the locations with the same semantic as the same semantic locations. Also, different users always engage in their activities at the same period in a cycle. We refer to the locations where different users engage in activities at the same time period as the same period locations. The above facts are the reason why the adversary can launch SLSA and why we define the two basic concepts. Note that Definition 2 can reflect activities in which a user visits some places periodically. Assume $r_{ij}$ and $r_{i'j}$ are the POIs where $u_j$ visits at the same period in different cycles and provide the same services. Then we can know that she visited the same location or participated in the same activity at the same time in two different periods.

## Adversary model

The principal goal of the adversary is to collect privacy information about a particular user by associating her real identity with the corresponding trajectories. In this work, we consider two types of adversaries. One is the unauthorized third party which could illegally obtain users' information by conducting eavesdropping attacks, purchasing from LBS, collecting from the released data. The other is the SPs which could obtain the current reviews sending by users and all historical original reviews. Additionally, they also could also know the identify information of all users. The reviews and identify information are stored on the server and can be seen by the SPs. The two types of adversaries are selfish and curious and infer visited and sensitive locations of each user using the gained data. In particular, the SPs will compromise with the unauthorized third party and sell users' reviews and identify information to them for self-interest. Hence, we consider the above two as the adversary in our paper. In our adversary model, we assume the adversary attempts to infer the following two types of trajectory privacy by using users' information.

- A particular user and her trajectories. The adversary analyses the spatial and temporal correlation of locations and reconstructs trajectories if a user's reviews are protected through pseudonym exchange. If so, she can know to whom the reconstructed trajectories belong.

- The most frequent semantic in the historical reviews. Based on the consistent lifestyle of a particular user, the adversary can count the most frequent semantic from historical data whose corresponding locations are most likely to be her real location.

## Motivation and basic idea

In existing LBSSs, a user submits her reviews to the LBSS server. The reviews and the user identifies are stored on the LBSS server as historical data and are visible to the SPs. To protect user privacy, a workable method is pseudonym. However, we cannot directly assign pseudonyms to users since the adversary knows users' real identities. So, one effective approach is to exchange reviews. But its weakness is that distorted trajectories always contain some sub-trajectories of original trajectories. In this paper, we refer to the trajectories before and after exchanging reviews as the original trajectory and the distorted trajectory, respectively. For the example in Table 1, $T_1$, $T_2$ and $T_3$ are original trajectories and the corresponding distorted trajectories are $T_1'$, $T_2'$ and $T_3'$, respectively. $a_2 \rightarrow a_3$ is a sub-trajectory of $T_1$. Adversaries can exploit the sub-trajectories to infer users' real identities. To illustrate this problem, we first give the following definition.

Definition 3 (sub-trajectory): For a trajectory $T_j$, we assume there exists a trajectory $T' = \{r_l | l = 1, 2, \cdots, m \text{ and } m \leq n\}$, where $r_l$ is the $l$-th location on $T'$ and $\forall r_l \in T'$, $r_l \in T_j$. Let $r_l$ and $r_{l+1}$ be equivalent to $r_{ij}$ and $r_{i'j}$ on $T_j$, respectively. For $\forall r_l, r_{l+1}$, if it satisfies the condition: $i < i'$, $T'$ is a sub-trajectory of $T_j$.

Definition 3 ensures that the consistency of spatial-temporal sequence of locations between a trajectory and its sub-trajectories, e.g., a user visits $r_l$ before $r_{l+1}$ in $T'$. A distorted trajectory can contain several sub-trajectories from different original trajectories.

**Table 1. Example of two trajectories exchanging reviews.**

| No.# | Original Trajectory | No.# | Distorted Trajectory |
|:---:|:---:|:---:|:---:|
| $T_1$ | $a_1 \rightarrow a_2 \rightarrow a_3 \rightarrow a_4$ | $T_1'$ | $a_1 \rightarrow b_2 \rightarrow b_3 \rightarrow c_4$ |
| $T_2$ | $b_1 \rightarrow b_2 \rightarrow b_3 \rightarrow b_4$ | $T_2'$ | $b_1 \rightarrow a_2 \rightarrow a_3 \rightarrow b_4$ |
| $T_3$ | $c_1 \rightarrow c_2 \rightarrow c_3 \rightarrow c_4$ | $T_3'$ | $c_1 \rightarrow c_2 \rightarrow c_3 \rightarrow a_4$ |

As stated in [9, 13], the adversary can still infer users' real identities if existing methods only encrypt user identities without distorting the original trajectory. Here, distorting refers to replace some locations on an original trajectory with some different locations. So, in the PR scenario, we need to distort the original trajectories after exchanging reviews. However, existing methods have not yet proposed a metric to measure how distorted the original trajectory is.

In our scenario, for a particular user, the adversary knows her trajectory stored on the LBSS server is a distorted trajectory. Yet, he wants to get her original trajectory by reconstructing the distorted trajectory. In many cases, an adversary can obtain a particular location or a sub-trajectory of a user. For example, the adversary may accidentally know Alice's home or path due to an encounter or walking together. Once obtaining these locations, he can exploit them to recover the original trajectory in a variety of ways, such as correlation attack [19], aggregated model [13]. Intuitively, for a distorted trajectory, the more locations the adversary knows, the more likely he is to recover the original trajectory. Note that the more unreplaced locations on an original trajectory, the more likely it is to be recovered. So, we uniformly use the maximal common sub-trajectory of a distorted trajectory and its corresponding original trajectory to represent locations that the adversary has already known.

According to the above analysis, the *distortion* metric is proposed to capture the correlation between the maximal common sub-trajectory and the distorted trajectory. It reflects how difficult it is for the adversary to recover the original trajectory. The larger the value of distortion is, the less likely the adversary is to recover the original trajectory. Then, we define the *distortion* metric as follows:

Definition 4 (*distortion*): For a particular user $u_j$, $T_j^o$ is his original trajectory. $T_j^d$ is the distorted trajectory of $u_j$. $T'(T_j^o, T_j^d)$ is the maximal common sub-trajectory of $T_j^o$ and $T_j^d$. We define the *distortion* between $T_j^o$ and $T_j^d$ as:

$$dis(T_j^o, T_j^d) = \frac{|T_j^o| - |T'(T_j^o, T_j^d)|}{|T_j^o|} \tag{1}$$

Where $|T'(T_j^o, T_j^d)|$ and $|T_j^o|$ are the number of locations of $T'(T_j^o, T_j^d)$ and $T_j^o$, respectively. $dis(T_j^o, T_j^d)$ denotes the probability that the adversary can recover complete $T_j^o$. Its physical meaning is that, for the $T_j^o$ with a fixed number of locations, the more replaced locations, the less likely the adversary can recover complete $T_j^o$.

Additionally, there is a threshold for the maximal common sub-trajectory. That is, the adversary can completely recover an original trajectory as long as he knows enough but not all locations on it. So, for $T_j^o$ and $T_j^d$, we must ensure that the distortion between them is bound by $\delta_j \in (0, 1]$. Note that we mainly consider users who have exchanged reviews with a particular user. $\delta_j = 0$ means two users did not exchange any reviews.

$$0 \leq dis(T_j^o, T_j^d) \leq \delta_j \tag{2}$$

In our scenario, the distorted trajectories of users who have exchanged reviews with $u_j$ also need to be bound by $\delta_j$. We assume $D_j$ is the set of these distorted trajectories. Then, for $\forall T_i^d \in D_j$, $T'(T_j^o, T_i^d)$ is the maximal common sub-trajectory of $T_j^o$ and $T_i^d$ and $T'(T_j^o, T_i^d) \neq \emptyset$, since $T_i^d$ contains some locations of $T_j^o$. It means that the adversary is likely to exploit $T_i^d$ to recover complete $T_j^o$. In particular, $T_j^o = T'(T_j^o, T_i^d)$ will allow the adversary to obtain all

locations of $T_j^o$. So, for $\forall T_i^d \in D_j$, we must ensure that the distortion between them is bound by $\delta_j$.

$$0 \leq dis(T_j^o, T_i^d) \leq \delta_j \tag{3}$$

The above analysis formalizes the conditions for satisfying trajectory privacy protection during exchanging reviews. If a distorted trajectory contains fewer exchanged locations, the adversary can exploit it to obtain the original trajectory. In this paper, we call this kind of attack review-based location correlation attack (RLCA).

Note that our paper mainly focuses on how the adversary exploits the sub-trajectory to obtain the original trajectory, rather than inferring which user the original trajectory belongs to. In other words, the adversary can obtain users' real identities once he determines the original trajectory.

As far as privacy protection is concerned, RLCA ignores the fact that a user always engages in the same activities periodically in the long term. Consider Alice who goes to some restaurants (perhaps not the same restaurants) near her workplace for lunch at 12:30 every day. The POIs that Alice visits will have the same semantics (called Food & Beverages Service). Though Alice exchanged reviews with others, using the historical data the adversary can still infer that Alice visited a place with the semantic 'Food & Beverages Service' since Alice appears more frequently than others and the POIs with the semantic 'Food & Beverages Service' appear more frequently than other POIs.

To clarify the above problem, we assume the adversary has obtained Alice's historical data during a period of time. In the historical data, Alice has submitted $n$ reviews to the LBSS server. For simplicity, suppose that $n$ POIs related to these reviews have the same semantics. For each review, we select other $k - 1$ users to form an anonymous group with Alice, in which $k$ users exchange their reviews and send them to the LBSS server. For these $k \times n$ POIs, there are $m$ different semantics denoted as $\{s_i | i = 1, 2, \ldots, m, 1 \leq m \leq ((k-1) \times n + 1)\}$ and $n_i$ is the number of $s_i$ appearing in these POIs. We assume $s_1$ is the semantic with which Alice has submitted reviews.

Then, the number of $s_1$ and other semantics appearing in these POIs are $k \times n - \sum_{i=2}^{m} n_i$ and $\sum_{i=2}^{m} n_i$, respectively. Among these $k \times n$ POIs, the frequency of $s_1$ is $p_1$, then:

$$p_i = \frac{k \times n - \sum_{i=2}^{m} n_i}{k \times n} \tag{4}$$

For $\forall s_i (2 \leq i \leq m)$, the frequency of $s_i$ is $p_i$, then:

$$p_i = \frac{n_i}{k \times n} \tag{5}$$

Consider that Alice sends her reviews in the long term. That is, Alice will send an unlimited number of reviews to the LBSS server. Then, we can get:

$$\begin{cases} \lim_{n \to \infty} p_i = \lim_{n \to \infty} (1 - \frac{\sum_{i=2}^{m} n_i}{k \times n}) = 1, (i = 1) \\ \\ \lim_{n \to \infty} p_i = \lim_{n \to \infty} (\frac{n_i}{k \times n}) = 0, (2 \leq i \leq m) \end{cases} \tag{6}$$

Furthermore, we denote these users as $\{u_l | l = 1, 2, \cdots, h, k \leq h \leq (k-1) \times n + 1\}$ and $n_l$ is the number of $u_l$ appearing in these users. Then, the number of Alice (We assume $u_1$ is Alice) appearing in these users is $n$. Except for Alice, the number of $u_l$ appearing among these users is

$n_l(1 \leq n_l \leq n)$. Then, we can get the frequency $q_l$ of $u_l$ as follows:

$$\begin{cases} q_l = \dfrac{n}{k \times n} = \dfrac{1}{k}, (l = 1) \\ q_l = \dfrac{n_l}{k \times n}, (2 \leq l \leq h) \end{cases} \tag{7}$$

When the reviews sent to the LBSS server are unlimited, Alice will appear in every review, others will not. So we can get the following formula:

$$\begin{cases} \lim_{n\to\infty} q_l = \lim_{n\to\infty}(\dfrac{n}{k \times n}) = \dfrac{1}{k}, (i = 1) \\ \lim_{n\to\infty} q_l = \lim_{n\to\infty}(\dfrac{n_l}{k \times n}) = 0, (2 \leq i \leq m) \end{cases} \tag{8}$$

According to the above formulas, we draw the following conclusions: (1) When $n$ tends to infinity, the frequency of $s_1$ and $s_i(i \neq 1)$ will be closed to 1 and 0, respectively. That is, so long as Alice submits large enough reviews with the same semantic to the LBSS server for a long term, the semantic of Alice's reviews must be much more frequent than every other semantic. (2) When the conditions are the same as (1), the frequency of Alice appearing in all users must be far more than others. By analyzing the historical data, the adversary can conclude that Alice and $s_1$ will appear in every review with extremely high probability. Once the adversary obtains some reviews including Alice and a location with $s_1$, it will be determined that Alice visited the location. In this paper, we call this kind of attack as semantic-based long-term statistical attack (SLSA).

The above analysis states two mechanisms through which the adversary launch RLCA and SLSA to obtain the trajectory when we protect user trajectory privacy by exchanging reviews in our scenario. (1) The adversary will recover a trajectory with enough unreplaced locations and know who it belongs to. (2) If a particular user periodically visits the POIs with the same semantic in the same time period for a long time, the frequency of the user and the frequency of the semantic will be much higher than those of other users in the historical data. Hence, our basic idea is that a user exchange reviews with different users as much as possible. Besides, the frequency difference of different semantics in the historical data is as small as possible.

To implement the above basic idea, our solution is to select users who exchange reviews from two aspects. First, before a user sends a review to the LBSS server each time, we try to select some other users to form an anonymous group. In the anonymous group, each user has at least one user whose the distortion between their trajectories does not exceed the threshold after they exchange reviews. It ensures that the adversary cannot recover the trajectories of every user in the anonymous group by launching RLCA. Second, we should select users to form anonymous groups and exchange reviews based on historical data. For a particular user, we select users to form an anonymous group, in which the frequency of each user and each semantic in the historical data are as the same as possible. It guarantees that the adversary cannot infer a user's location by launching SLSA.

## System architecture and algorithm design

### System architecture

To select suitable users who exchange reviews to resist RLCA and SLSA, our system architecture should consider two facts: (1) the SPs are adversaries and we cannot storage non-anonymous user historical data anonymized on the LBSS server; (2) the overhead of storage and

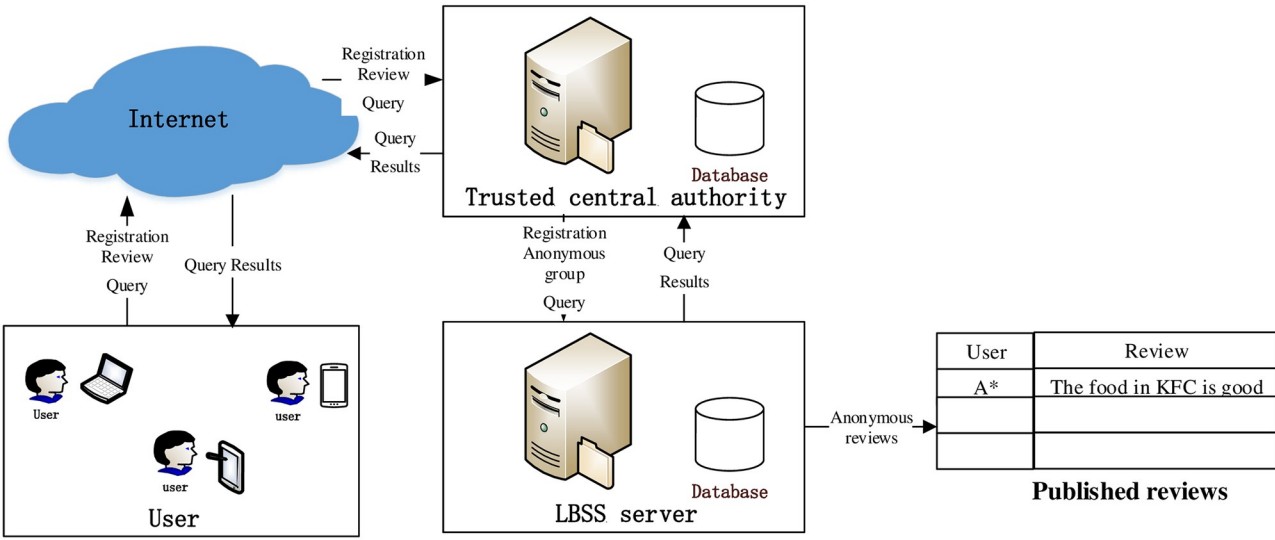

**Fig 2. The system architecture.**

calculation are huge and we cannot implement them on mobile terminals. Therefore, we employcentralized architecture as our system architecture. Our system architecture contain three roles as shown in Fig 2.

Users: In our system, users can use microcomputers, mobile terminals, etc., to register with the Trusted Central Authority (TCA) by sending a registration request. Users can also send query requests, reviews, etc., to TCA so that they can query and review the services provided by POIs.

TTP: TTP is an independent and trusted third-party server, which receives query requests and registration requests from users and forwards them to the LBSS server. It receives query results from the LBSS server and returns them to users. During the registration process, TTP stores users' real identities information. Additionally, the TTP server stores the user reviews in the database, and selects users to exchange reviews to protect user privacy, also stores some data related to privacy protection functions, such as POIs within some cities and their services.

LBSS server: It provides users with services such as query, registration, and review. Specifically, the LBSS server receives query requests and registration requests from the TTP server and returns the query results to it. The LBSS server stores users' real identities information and the anonymized reviews in the database. The LBSS server also publishes reviews on the Internet.

## The algorithm framework

In an LBSS, users not only wish to enjoy the business services, but also hope to publish objective reviews for the service so that others can also enjoy them. If user $u_j$ wants to publish reviews, she needs to register with the system by sending her real identity. However, her trajectory privacy is inevitably leaked, since the SPs are untrustworthy and her identity information and reviews are stored on the server. Hence, we propose a method to focus on how to select users who exchange reviews. To select proper users, the TTP server first select $u_j$ and other $k − 1$ users to form an anonymous group and each selected user exchange reviews with another user by running algorithm 1. If the trajectories of $k$ users cannot resist RLCA after exchanging

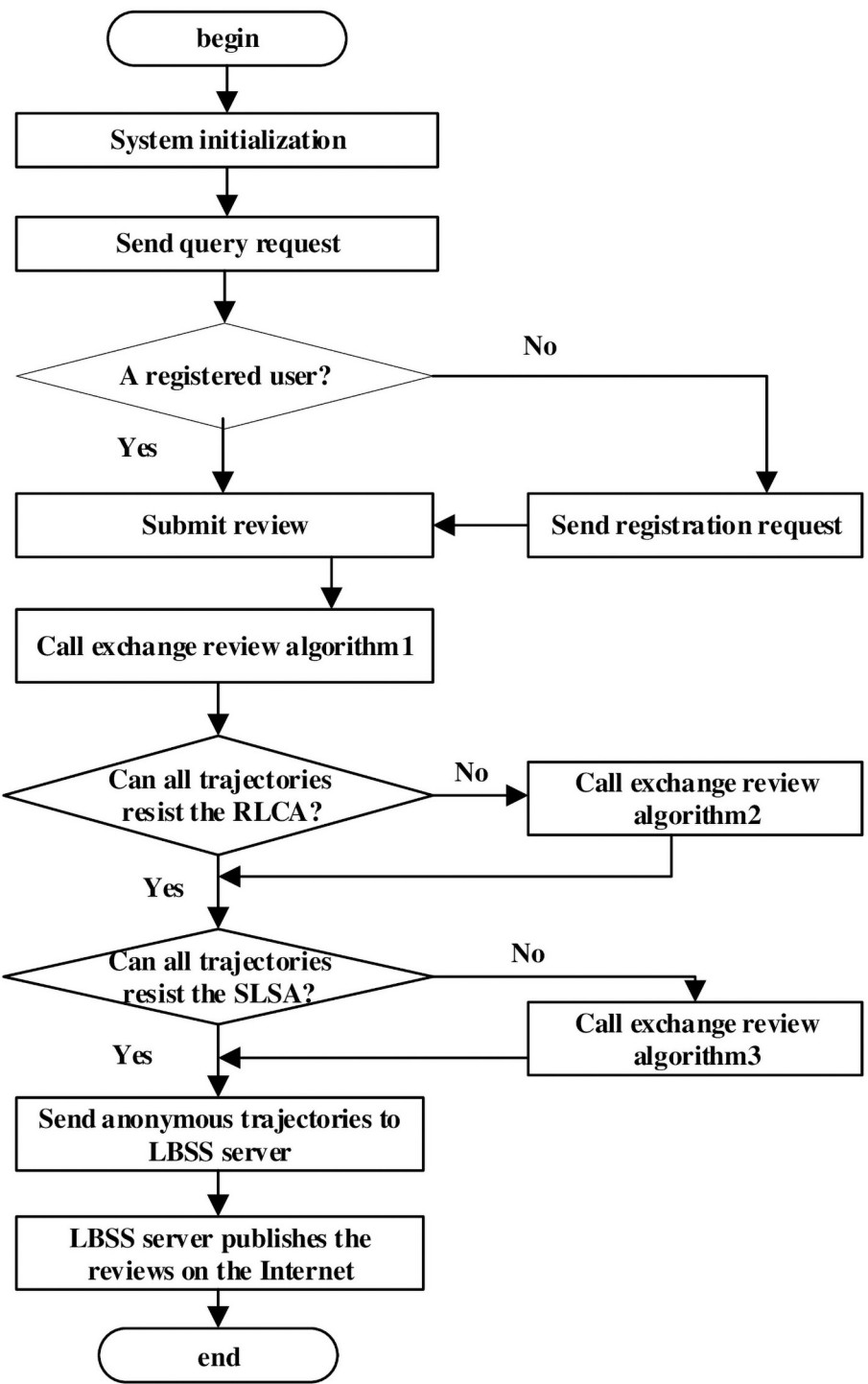

**Fig 3. The framework of our algorithms.**

reviews, the TTP server needs to reselect users to exchange reviews by running algorithm 2; if the trajectories of $k$ users can resist RLCA but not SLSA, the TTP server needs to reselect users to exchange reviews by running algorithm 3; then, the TTP server send the trajectories that can resist RLCA and SLSA. The framework of our algorithms is shown in Fig 3.

## The algorithm design

**RUS algorithm.** The main purpose of RUS algorithm is to randomly select users to exchange reviews without considering RLCA and SLSA. To better protect the trajectory privacy, every time a user submits a review, we select $k-1$ users whose reviews have not been exchanged for her to form an anonymous group, in which each user selects another one to exchange review. Let $O = \{T_1^o, T_2^o, \cdots, T_N^o\}$ and $A = \{T_1^d, T_2^d, \cdots, T_N^d\}$ denote the set of original trajectories and the set of distorted trajectories already stored on the TTP server, respectively. For $u_j$, each location on $T_j^o$ corresponds to a review and has been used to exchange reviews with other users. For each location on $T_j^d$, it corresponds to the review that has been used to exchange with one review of $u_j$. Suppose $u_j$ submits a review for a POI that needs to be exchanged to the TTP server and the location corresponding to the POI is denoted as $r_j'$. Only after the review of $u_j$ is exchanged can $r_j'$ be added to $T_j^o$. If $u_j$ and $u_i$ exchange reviews, $r_i'$ will replace $r_j'$ and is added to $T_j^d$. Here, we denote $r_i'$ on $T_j^d$ as $R(r_i' \leftarrow r_j')$.

In an LBSS, every time a user submits a review, RUS algorithm needs to search for all unexchanged reviews received by the TTP server. According to the given security parameter $k$, we select $k$ users to form an anonymous group and exchange reviews. The process to solve this problem is shown below.

First, when user $u_j$ submits a review, RUS algorithm needs to input all unexchanged reviews and get the set of locations $R = \{r_j', r_{p_1}', r_{p_2}', \cdots, r_{p_L}'\}$ that correspond to these reviews. Where $r_{p_l}' \in R$ is the location of $u_{p_l}$ and $t(r_{p_l}')$ is the time when $u_{p_l}$ has reviewed $r_{p_l}'$. The system also needs to determine the security parameter $k$ (in our paper, $k$ is a constant not less than 3). Given $k$, RUS algorithm needs to select $u_j$ and other $k-1$ users to form an anonymous group. A bigger $k$ leads to more users who can exchange reviews with $u_j$ and better trajectory privacy protection.

Second, $u_j$ and other $k-1$ users are selected and their reviews are exchanged according to R. At the beginning of RUS algorithm, we first get the set of locations $R'$ by sorting the locations in $R$ in chronological order in which these locations are reviewed. Assume $R = \{r_j', r_{p_1}', r_{p_2}', r_{p_3}'\}$. If $t(r_{p_1}') < t(r_{p_3}') < t(r_{p_2}')$, then $R' = \{r_j', r_{p_1}', r_{p_3}', r_{p_2}'\}$. In $R'$, we first select the location $r_{p_l}'$ with the smallest time interval for $r_j'$. If $r_{p_l}'$ does not satisfy at least one condition: (1) $POI(r_{p_l}') \neq POI(r_j')$; (2) $\tau(r_{p_l}') = \tau(r_j')$, we continue to select the location $r_{p_{l+1}}'$ with the second smallest interval for $r_j'$ until $r_{p_{l+1}}'$ meets both conditions. Then, $u_j$ and $u_{p_{l+1}}$ exchange locations. As far as $u_{p_{l+1}}$ and her exchanged location $r_j'$ is concerned, we select a location for $r_j'$ to exchange as we do for $u_j$. At last, we follow the above steps to select $k$ users including $u_j$ to form an anonymous group $G$. In $G$, each element is a two-tuple composed of the user and the exchanged location.

Third, RUS algorithm searches the database stored on the TTP server to find $O'$ and $A'$, which are the subset of $O$ and the subset of $A$, respectively. $O'$ and $A'$ respectively contain the original trajectory and the distorted trajectory of $k$ users in $G$. For each location in $G$, we add the location to its corresponding user. For example, for $r_{p_l}'$, we add it to $T_{p_l}^o$, and then add it $T_j^d$ if $u_{p_l}$ and $u_j$ exchange their reviews.

Finally, we output the anonymous group contains $k$ users, their original trajectories and distorted trajectories. The anonymous group is sent to the LBSS server, and the original trajectories and the distorted trajectories are stored in the database on the TTP server. The pseudocode is described as Algorithm 1.

Algorithm 1 describes how RUS algorithm selects users and exchanges reviews. It ensures that the SPs can obtain a user's real identity and real review, but don't know who submits the

review. Therefore, it can effectively protect users' trajectory privacy. However, when selecting users to exchange reviews, RUS algorithm fails to consider RLCA and how $k$ will lead to the leakage of trajectory privacy after exchanging reviews. As a result, we should enhance RUS algorithm so that it can address the problem.

**Algorithm 1**: Random-User Selection Algorithm

**Input:** all reviews that have not been exchanged with others, security parameter $k$

**Output:** the anonymous group $G$ which contains $k$ users and their reviews that have been exchanged with others, the set of the original trajectory $O'$, the set of distorted trajectory $A'$.

1 Get the set of locations $R = \{r'_j, r'_{p_1}, r'_{p_2}, \cdots, r'_{p_L}\}$ from all reviews;

2 $R' \leftarrow$ sort the locations in $R$ in chronological order when the locations are reviewed;

3 $G \leftarrow \emptyset$, $O' \leftarrow \emptyset$, $A' \leftarrow \emptyset$, initial user $temp_u = u_j$, $R' = R'\backslash\{r'_j\}$;

4 **while** $number(G) \leq k$ **do**

5 **for** $r'_{p_l} \in R'$ **do**

6 **if** $POI(r'_{p_l}) \neq POI(r'_j)$ and $\tau(r'_{p_l}) = \tau(r'_j)$ **then**

7 $temp_u$ and $u_{p_l}$ exchange locations $r'_j$ and $r'_{p_l}$;

8 $G \leftarrow (temp_u, r'_{p_l})$; //$(temp_u, r'_{p_l})$ means the location of $temp_u$ is $r'_{p_l}$ after exchanging review;

9 $R' = R'\backslash\{r'_{p_l}\}$, $temp_u = u_{p_l}$;

10 Searches the database stored on the TTP server and get the original trajectories and the distorted trajectories of $k$ users in $G$;

11 $O' \leftarrow$ add each location in $G$ to the corresponding original trajectory;

12 $A' \leftarrow$ add each location in $G$ to the corresponding distorted trajectory;

13 **Return** $G$, $O'$, $A'$

**USR-RLCA algorithm.** To protect trajectory privacy more effectively, RUS algorithm must be enhanced by considering the RLCA and $k$. According to formulas Eqs (2) and (3), to resist RLCA, the distortion between an original trajectory and any distorted trajectories exchanged reviews with the original trajectory must be less than $\delta_j$. It means a user should avoid exchanging reviews with the same user multiple times as much as possible. In other words, a user should exchange reviews with as many users as possible. However, more users always lead to higher overload due to the selection of more users.

Another problem to be considered is, for an original trajectory, no matter how many reviews of it are exchanged, the adversary can always recover it. To illustrate it, we assume an original trajectory $T^o_j$ contains 10 locations and $\delta_j = 0.4$. It means that any user can exchange reviews with $u_j$ no more than 4 times. However, if only two trajectories, no matter how many times they exchange reviews, the adversary can still infer $T^o_j$. To address the problem, each trajectory (including $T^d_j$) should contain at most 4 identical locations with $T^o_j$ after exchanging reviews. At this time, during the exchange of all reviews, $u_j$ should exchange reviews with at least $\lfloor 10/(0.4 * 10)\rfloor + 1 = 3$ users. That's, $k$ is determined by $\delta_j$ and the number of locations on $T^o_j$. To address this problem, every time a user exchanges reviews with others, we set the number of users in the anonymous group to be as least $\lfloor\frac{1}{\delta_j}\rfloor + 1$. So, $k$ is represented as

$$k \geq \lfloor\frac{1}{\delta_j}\rfloor + 1 \tag{9}$$

Based on the aforementioned analysis, we propose USR-RLCA algorithm to select users and exchange reviews. USR-RLCA is an enhancement RUS algorithm, since it considers the

threshold $\delta_j$ and $k$. By running USR-RLCA, $u_j$ submits a review each time, at least $\lfloor\frac{1}{\delta_j}\rfloor + 1$ users are selected to form an anonymous group and the *distortion* between $T_j^o$ and any distorted trajectories is ensured to be less than $\delta_j$.

Algorithm 2 gives the pseudo-code to describe how USR-RLCA algorithm selects users and exchanges reviews. When receiving the reviews submitted by $u_j$, the TTP server first passes the parameters set and the unexchanged review to USR-RLCA algorithm. USR-RLCA gets the locations set $R = \{r_j', r_{p_1}', r_{p_2}', \cdots, r_{p_L}'\}$ and sorts it into the locations set $R'$ in chronological order in which these locations are reviewed by following the method in lines 1 to 2 of RUS algorithm. Then, we construct the distorted trajectories set which can resist RLCA by selecting $k - 1$ users to exchange reviews for $u_j$. We first search the database stored on the TTP server and get $T_j^o$ and $T_j^d$, which are the original trajectory and the distorted trajectory of $u_j$ after the last exchange of a review. For the review of $u_j$ this time, in line 3, we denote the distorted trajectory variable as $T_{dis}^d$. $T_j^d$ is the initial value of $T_{dis}^d$. At the same time, $T_j^o$ has become a new original trajectory $T_j^o \bigcup \{r_j'\}$ after adding the location $r_j'$. Then, we select users to form an anonymous group where the distortion between $T_j^o \bigcup \{r_j'\}$ and each user's distorted trajectory added to $r_{p_l}'$ is less than $\delta_j$ (line 6 ~16). We first choose $r_{p_1}', r_{p_2}', \cdots$ for $u_j$. For $u_{p_l}$, if $r_{p_l}'$ satisfies three conditions: (1) $POI(r_{p_l}') \neq POI(r_j')$; (2) $\tau(r_{p_1}') = \tau(r_j')$; (3) $\frac{|T_j^o| - \left|T'(T_j^o, T_{dis}^d \bigcup \{r_{p_l}'\})\right|}{|T_j^o|} \in (0, \delta_j]$, $u_j$ and $u_{p_l}$ can exchange their reviews and $T_{dis}^d \cup \{r_{p_l}'\}$ is added to the set of distorted trajectory $A'$. Then, we set the user variable $temp_u$ as $u_{p_l}$ and $T_{dis}^d$ as $T_{dis}^d \bigcup \{r_j'\}$ to repeat the steps until there are k distorted trajectories in $A'$. At last, we follow the method in lines 1 to 2 of RUS algorithm to output the anonymous group G, the original trajectories set $O'$ and the distorted trajectories set $A'$ (line 17).

**USR-SLSA algorithm.**   Considering SLSA, during the original trajectories being exchanged, USR-RLCA algorithm needs to ensure that the difference of different semantics and the difference of different users probability are as small as possible.

**Algorithm 2**: User Selection to Resist RLCA Algorithm

```
Input: all reviews that have not been exchanged with others, security
parameter k, δⱼ
Output: the anonymous group G which contains k users and their reviews
that have been exchanged with others, the set of the original trajec-
tory O′, the set of distorted trajectory A′.
```
1 Get the set of locations $R = \{r_j', r_{p_1}', r_{p_2}', \cdots, r_{p_L}'\}$ from all reviews;
2 $R' \leftarrow$ sort the locations in $R$ in chronological order when the locations are reviewed;
3 $G \leftarrow \emptyset$, $O' \leftarrow \emptyset$, $A' \leftarrow \emptyset$, initial user $temp_u = u_j$, initial distorted trajectory $T_{dis}^d = T_j^d$, $R' = R' \backslash \{r_j'\}$;
4 $T_j^o \leftarrow T_j^o \bigcup \{r_j'\}$;
5 $O' \leftarrow T_j^o$;
6 **while** $number(G) \leq k$ **do**
7   **for** $r_{p_l}' \in R'$ **do**
8     **if** $POI(r_{p_l}') \neq POI(r_j')$ *and* $\tau(r_{p_l}') = \tau(r_j')$ *and* $\frac{|T_j^o| - \left|T'(T_j^o, T_{dis}^d \bigcup \{r_{p_l}'\})\right|}{|T_j^o|} \in (0, \delta_j]$ **then**
9       Searches the database stored on the TTP server and get $T_{p_l}^o$ and $T_{dis}^d$;
10         $T_{p_l}^o \leftarrow T_{p_l}^o \bigcup \{r_{p_l}'\}$;
11         $temp_u$ and $u_{p_l}$ exchange locations $r_j'$ and $r_{p_l}'$;

12  $T_{dis}^d \leftarrow T_{dis}^d \bigcup \{r'_{p_l}\}$;

13  $A' \leftarrow T_{dis}^d$,  $G \leftarrow (temp_u, r'_{p_l})$,  $O' \leftarrow T_{p_l}^o$;

14  $R' = R' \backslash \{r'_{p_l}\}$,  $temp_u = u_{p_l}$,  $T_{dis}^d \leftarrow T_{p_l}^d$;

15 **Return** $G$, $O'$, $A'$

For the one hand, to prevent the adversary from launching SLSA, we consider the case that $u_j$ sends $n$ reviews in a period of time and selects $k-1$ users to exchange reviews each time. Obviously, $u_j$ will participate in exchanging reviews every time while others are not, because they may not visit the POIs at the same time as $u_j$ or be selected by USR-RLCA algorithm to participate in exchanging reviews. Therefore, there are two solutions to this problem. An optimal solution is that we select the same users to form an anonymous group every time $u_j$ sends a review. However, since it is impossible to ensure that each user and $u_j$ submit reviews at the same time, the optimal solution is not feasible, especially $u_j$ submit a large number of reviews in a long term. The other solution is that we can select different users but ensure that the difference between the probability of any user (denote the set of all users as $D_u = \{u_{a_1}, u_{a_2}, \cdots, u_{a_D}\}$) and $u_j$ is bound by the threshold $\delta_u$ in the long term. Then, for each user $u_{a_d} \in D_u$, the solution can be formalized as Eq (10).

$$d_u(u_{a_d}, u_j) = |p(u_{a_d}) - p(u_j)| \leq \delta_u \qquad (10)$$

For the other hand, the other question for $u_j$ is that she will submit reviews with the same semantic (denote it as $s_j$) during the same time period in each cycle while other users selected to exchange reviews are not. This causes that the number of semantic $s_j$ is far more than other semantics during the same time period of each cycle in the long term. In other words, the $p(s_j)$ (probability of $s_j$) is much bigger than the probability of other semantics. According to the analysis that $u_j$ has the highest probability among all users and $s_j$ also has the highest probability among all semantics, the adversary can refer that $u_j$ is the user who visits a POI with $s_j$. So, for $s_j$ and all semantics $S = \{s_1, s_2, \cdots, s_S\}$, we ensure that the difference between the probability of $s_j$ and $\forall s_i \in S$ is bound by the threshold $\delta_s$.

$$d_s(s_i, s_j) = |p(s_i) - p(s_j)| \leq \delta_s \qquad (11)$$

**Algorithm 3**: User Selection to Resist SLSA Algorithm

```
Input: all reviews that have not been exchanged with others, security
parameter k, δ_j, δ_u, δ_s
Output: the anonymous group G which contains k users and their reviews
that have been exchanged with others, the set of the original trajec-
tory O', the set of distorted trajectory A'.
1 Get the set of locations R = {r'_j, r'_{p_1}, r'_{p_2}, ···, r'_{p_L}} from all reviews;
2 R' ← sort the locations in R in chronological order when the loca-
tions are reviewed;
3 Search the distorted trajectories of u_j and get the set of locations
R'' which are at the same time period with r'_j;
4 According to R'', get the set of semantics S = {s_1, s_2, ···, s_S} and the
set of users U' = {u'_1, u'_2, ···, u'_U};
5 Count the number of locations with different semantics and the num-
ber of different users in R'';
6 G ← ∅, O← ∅, A' ← ∅, R''' ← R'', initial user temp_u = u_j, initial dis-
torted trajectory T_dis^d = T_j^d, R' = R'\{r'_j};
7 T_j^o ← T_j^o ∪{r'_j};
8 O' ← T_j^o;
9 while number(G)≤k do
10   for r'_{p_l} ∈ R' do
```

```
11        if s_{p_l} ∈ S and u_{p_l} ∈ U'  then
12           Compute p(s_{p_l}) in R''⋃{s_{p_l}} and p(u_{p_l}) in U'⋃{u_{p_l}};
13           if there exists at least one s_i ∈ S\{s_{p_l}} which d_s(s_i,s_{p_l}) ≥ δ_s  or
u'_l ∈ U'\{u_{p_l}} which d_u(s_i,s_{p_l}) > δ_u  then
14              exit
```

$$
15 \qquad \textbf{else if } POI(r'_{p_l}) \neq POI(r'_j) \ and \ \tau(r'_{p_l}) = \tau(r'_j) \ and \ \frac{|T^o_j| - \left| T'(T^o_j, T^d_{dis} \bigcup \{r'_{p_l}\}) \right|}{|T^o_j|} \in (0, \delta_j]
$$

```
then
16              Searches the database stored on the TTP server and get T^o_{p_l} and
T^d_{dis};
17              T^o_{p_l} ← T^o_{p_l}⋃{r'_{p_l}};
18              temp_u and u_{p_l} exchange locations r'_j and r'_{p_l};
19              T^d_{dis} ← T^d_{dis}⋃{r'_{p_l}};
20              A' ← T^d_{dis},  G ← (temp_u, r'_{p_l}),  O' ← T^o_{p_l};
21              R' = R'\{r'_{p_l}},  temp_u = u_{p_l},  T^d_{dis} ← T^d_{p_l};
22    if R' = ∅ and number(G)<k then
23      for r'_{p_l} ∈ R''' do
24        if s_{p_l}∉S or u_{p_l}∉U' then
25          Repeat lines 12 to 22
26 Return G, O', A'
```

The Eq (11) makes it impossible for the adversary to get the inference that the POI with the semantics $s_j$ is the most likely to visit. Combined with the formula (10), the adversary fails to establish the connection between $u_j$ and the location with the semantics $s_j$. Based on the aforementioned formulas, we propose USR-SLSA algorithm to select users to exchange reviews such that it can resist the SLSA.

Algorithm 3 gives the pseudo-code to describe how USR-SLSA algorithm selects users and exchanges reviews. First, when receiving the reviews submitted by $u_j$, the TTP server gives $k$, $\delta_j$, $\delta_u$, $\delta_s$ to USR-SLSA algorithm. The locations set $R = \{r'_j, r'_{p_1}, r'_{p_2}, \cdots, r'_{p_L}\}$ should also be given and sorted in chronological order by following the method in lines 1 to 2 of USR-RLCA algorithm. Considering that we will statistic the probabilities of different users and different semantics in the distorted trajectories of $u_j$ in a long term, we also need to collect the same semantic locations and the same period locations of $r'_j$. So, we search the distorted trajectories of $u_j$ on the TTP server and get the locations set $R''$ containing the two types of locations in line 3. After obtaining $R''$, we calculate the probabilities of different users $U' = \{u'_1, u'_2, \cdots, u'_U\}$ and the the probabilities of different semantics $S = \{s_1, s_2, \cdots, s_S\}$ in it. Second, USR-SLSA algorithm selects users to form an anonymous group. This step aims to select locations and users that can meet formulae Eqs (10) and (11) for $r'_j$ to exchange such that USR-SLSA algorithm can resist SLSA. For every review of $u_j$, the optimal solution is that every anonymous group contains the same users and the same semantics. So, we prefer to select these locations where the corresponding users belong to $U'$ and semantics belong to S. In the first step, we traverse $R'$. For $r'_{p_l} \in R'$, if its corresponding user $u_{p_l}$ and all users in $U'$ satisfy Eq (10) and its corresponding semantic $s_{p_l}$ and all semantics in $S$ satisfy the Eq (11) (line 11 to 13) and $r'_{p_l}$ satisfies

three conditions: (1) $POI(r'_{p_l}) \neq POI(r'_j)$; (2) $\tau(r'_{p_l}) = \tau(r'_j)$; (3) $\frac{|T^o_j| - \left| T'(T^o_j, T^d_{dis} \bigcup \{r'_{p_l}\}) \right|}{|T^o_j|} \in (0, \delta_j]$

(line 15), $r'_{p_l}$ and $u_{p_l}$ can be select to form the anonymous group. Then, we get $G$, $O'$, $A'$ by following the method in lines 9 to 14 of USR-SLSA algorithm (line 16 to 21). In the second step, when traversing $R'$, if the users in the anonymous group are less than $k$, we traverse locations

in $R'''$. For a location $r'_{p_l} \in R'''$, if its corresponding user $u_{p_l}$ and all users in $U'$ satisfy Eq (10) or its corresponding semantic $s_{p_l}$ and all semantics in $S$ satisfy Eq (11) (line 24 to 26), we implement the process in lines 12 to 22 for $r'_{p_l}$ and finally get $G$, $O'$, $A'$ (line 27). At last, output the anonymous group $G$, the original trajectories set $O'$ and the distorted trajectories set $A'$ (line 29).

## Feasibility discussion

In this section, we discuss the feasibility of the proposed scheme in terms of both implementation and security. Specifically, following the aforementioned goals, we discuss whether our scheme can be implemented and achieve the desired privacy protection requirements.

### Implementation analysis

**Users and System Providers (SPs).** The core of our scheme is that users exchange reviews with each other. It means that for Alice, in a public review list, Bob will publish her review. Therefore, the first question we consider in our implementation analysis is whether users are willing to exchange reviews with others.

For LBSSs, users' identities are anonymized in various ways, such as pseudonyms, hiding key characters, etc. It indicates that the user is more concerned about the impact of the review on the business than about who published it. In fact, by storing the original trajectory in a database, TTP servers can still maintain authentic review lists for each user and display the review list in a way that is personally visible to each user. Therefore, it is feasible to assume that users are willing to exchange reviews to protect the trajectory privacy.

For the SPs, we mainly consider whether they are willing for users to exchange reviews with each other when it is legally regulated. In general, the SPs are motivated by the desire for users to submit as many authentic reviews as possible so that they can build an objective reputation for the business. In our scenario, although users can exchange reviews, they do not submit dummy reviews as the $k$ anonymous. Therefore, it does not affect the objectivity of the business's reputation. At the same time, considering that trajectory privacy can be protected, users will be willing to submit much more reviews. So, it is feasible to assume that the SPs are willing for users to exchange reviews.

**The existence of the solution.** In our scenario, the ideal solution of our scheme is that we can select $k - 1$ users for each review of user $u_j$ to form an anonymous group to exchange reviews and that the trajectories of all users in the anonymous group cannot be identified by the adversary exploiting RLCA and SLSA. However, as the aforementioned analysis in Section USR-RLCA algorithm, such ideal solution does not always exist. Thus, we prove that our solution is feasible by demonstrating the existence of such an ideal solution in this section. Let $R = \{r'_j, r'_{p_1}, r'_{p_2}, \cdots, r'_{p_L}\}$ be the locations corresponding to the reviews of users that have not been exchanged with others. Where $r'_j$ is the location of $u_j$. $T^o = \{T^o_j, T^o_{p_1}, T^o_{p_2}, \cdots, T^o_{p_L}\}$ and $T^d_j = \{T^d_{p_1}, T^d_{p_2}, \cdots, T^d_{p_L}\}$ are the set of original trajectories and the distorted trajectories corresponding to the locations in $R$. $U = \{u_j, u_{p_1}, u_{p_2}, \cdots, u_{p_L}\}$ is the set of users corresponding to the locations in $R$. We first give the following definition.

Definition 5: For any user in $U$, e.g., $u_j$, the solution of our scheme exists if we can select $k - 1$ users from $U$ to achieve the goal of our scheme by exchanging reviews with each other.

In this paper, our scheme achieves three goals of trajectory privacy protection, i.e., randomly selecting users to exchange reviews, resisting RLCA, and resisting SLSA, which is achieved by running the algorithms of RUS, USR-RLCA and USR-SLSA, respectively.

Theorem 1: For our scheme, the solution exists.

Proof: We consider the solutions of three algorithms of our scheme from the following aspects.

RUS algorithm: The existence of a solution to RUS algorithm refers that it can select $k − 1$ users from $U$ to exchange reviews for $u_j$. Obviously, it can easily achieve this goal, since the number of users in $U$ is greater than $k − 1$.

USR-RLCA algorithm: Given parameters $k$ and $\delta_j$, the existence of a solution to USR-RLCA algorithm refers that it can select $k − 1$ users from $U$ and ensure that each of these $k$ trajectory (contain $T_j^d$) contains at most $(|T_j^d| + 1) \times \delta_j$. Using Eq (9) we have

$$\frac{|T_j^d| + 1}{k} \leq \frac{|T_j^d| + 1}{\lfloor \frac{1}{\delta_j} \rfloor + 1} < \frac{|T_j^d| + 1}{\lfloor \frac{1}{\delta_j} \rfloor} \leq (|T_j^d| + 1) \times \delta_j$$

Therefore, USR-RLCA algorithm can select $k − 1$ users from $U$ that achieves its goal to exchange reviews.

USR-SLSA algorithm: Given parameters $k$, $\delta_j$, $\delta_u$ and $\delta_s$, the existence of a solution to USR-SLSA algorithm refers that it can select $k − 1$ users from $U$ to exchange reviews and ensure that the difference in probability between different semantics or the probability of $k$ users satisfies Eqs (10) and (11). For every review of $u_j$, if the selected $k$ users are the same or the locations of the selected $k$ users have the same semantics, it will satisfy Eq (11). USR-SLSA algorithm can easily select such $k$ users whose locations have the same semantics every review of $u_j$, since $R$ contains enough locations. Besides, USR-SLSA algorithm can also select the same users every review of $u_j$. Hence, USR-RLCA algorithm can select $k − 1$ users from $U$ that achieves the goal to resist SLSA.

**Time complexity.** Our scheme consists of three algorithms of RUS, USR-RLCA, and USR-RLCA. RUS algorithm includes of two processes of sorting the set $R$ by the order of time and selecting $k$ users to form an anonymous group. Assume $R$ contains $L$ locations (except $u_j$). In the worst case, the time complexity of sorting the set $R$ is $O(L^2)$. In the process of selecting $k − 1$ users for $u_j$ from $R'$, in the worst case, the time complexity is $O(\sum_{i=1}^{k}(L − (k − 1)))$. Therefore, the time complexity of RUS algorithm in the worst case is $O(L^2 + \sum_{i=1}^{k}(L − (k − 1)))$. USR-RLCA algorithm contains the same processes as RUS algorithm. The difference is that in the process of selecting $k − 1$ users, USR-RLCA needs to calculate the distortion between the two trajectories. Assume $T^o = \{T_j^o, T_{p_1}^o, T_{p_2}^o, \cdots, T_{p_L}^o\}$ is the set of original trajectories and $T^d = \{T_j^d, T_{p_1}^d, T_{p_2}^d, \cdots, T_{p_L}^d\}$ is the set of distorted trajectories before $r_j'$ is exchanged. For this process, since computing the sub-trajectory of an original trajectory and a distorted trajectory consumes most of the computational resources. The time complexity is $O((|T_j^o| + 1) \times (|T_i^d| + 1))$. Hence, the time complexity of USR-RLCA algorithm in the worst case is $O(L^2 + \sum_{i=1}^{k}(L − (k − 1)) + (|T_j^o| + 1) \times (|T_i^d| + 1))$.

USR-SLSA algorithm also contains the same processes as RUS algorithm. But USR-SLSA algorithm needs to prioritize these locations where the users and the semantics are the same as the preceding anonymous groups. Let $U' = \{u_1', u_2', \cdots, u_U'\}$ and $S = \{s_1, s_2, \cdots, s_S\}$ denote the users and semantics in the preceding anonymous groups. Assume there are $k_0$ locations where the corresponding users belong to $U'$ and semantics belong to $S$. Then, there are $k − k_0$ locations where the corresponding users belong to $U'$ or semantics belong to $S$. For these $k_0$ locations, the time complexity (denote as $O_{k'}$) is $O_{k'} = O(\sum_{i=1}^{k_0}(L − (k_0 − 1)) \times (U − (k_0 − 1)) \times (S − (k_0 − 1)) + (|T_j^o| + 1) \times (|T_i^d| + 1))$. For these $k − k_0$ locations, assume there are $k_u$ locations where corresponding users belong to $U'$ but semantics don't belong to $S$ and $k_s$ locations where corresponding users belong to $S$ but users don't belong to $U'$. Then, the time complexity

(denote as $O_{k''}$) is $O_{k''} = O(\sum_{i=1}^{k_u}(L - k_0 - (k_u - 1)) \times (U - k_0 - (k_u - 1)) + (|T_j^o| + 1) \times (|T_i^d| + 1) + \sum_{i=1}^{k_s}(L - k_0 - (k_s - 1)) \times (U - k_0 - (k_s - 1)) + (|T_j^o| + 1))$. Hence, the time complexity of USR- SLSA algorithm in the worst case is $O(L^2) + O_{k'} + O_{k''}$.

## Security analysis

In our scenario, the adversary gaining trajectory privacy means that the adversary infers an original trajectory and the user to whom it belongs. In our scheme, the adversary gains trajectory privacy in three ways: (1) There is a correspondence between a user and her trajectory; (2) The adversary can infer the original trajectory by launching RLCA; (3) The adversary can infer the POI that a user periodically visits by launching SLSA. For (1), it is clear that there is no correspondence between the user and her trajectory by adopting our scheme. Hence, in this section, we only prove that our scheme can resist both RLCA and SLSA.

1. Resisting to RLCA. In this part of the analysis, the adversary knows some locations that a user has visited. Once he finds that a distorted trajectory of her original trajectory contains some of these locations, he will likely infer her original trajectory.

Definition 6: For $T_j^o$ and the corresponding distorted trajectories $T^d = \{T_1^d, T_2^d, T_3^d, \cdots, T_k^d\}$, our scheme can resist RLCA if the distortion between $T_j^o$ and each distorted trajectory in $T^d$ is less than the threshold $\delta_j$.

Theorem 2: Our scheme is resistant to RLCA.

Proof: For each distorted trajectory in $T^d$, USR-RLCA algorithm and USR-RLCA algorithm compute the distortion between $T_j^o$ and it. Only the distortion between them is bounded by (0, $\delta_j$], it can be the distorted trajectory of $T_j^o$ and be added to $T^d$. Therefore, the two algorithms ensure that our scheme can resist RLCA.

2. Resisting to SLSA. As the aforementioned analysis in USR-SLSA algorithm, when the LBSS server accepts the anonymous group $G$ from the TTP server which is formed for $u_j$ to exchange reviews with others, the adversary can know all probabilities of users $U' = [u_j, u_1, u_2, \cdots, u_{k-1}]$ and semantics $S = \{s_j, s_1, s_2, \cdots, s_k'\}$. He also knows the difference in probability between $s_j$ and other semantics and the difference in probability between $u_j$ and other users. For $\forall s_i \in S$ and $\forall u_l \in U'$, once the $d_s(s_i, s_j)$ is more than $\delta_s$, and the $d_u(u_l, u_j)$ is more than $\delta_u$, he will infer that the review with the semantics $s_j$ is the most likely one $u_j$ has visited.

Definition 7: For $u_j$ and the semantic $s_j$, our scheme can resist SLSA if each $u_l \in U'$ and the semantic $s_i \in S$ corresponding to the location of $u_l$ satisfy one of the two following conditions: i) $d_s(s_i, s_j)$ are less than the threshold $\delta_s$ and all $d_u(u_l, u_j)$ are less than the threshold $\delta_u$; ii) $d_s(s_i, s_j)$ are less than the threshold $\delta_s$ or all $d_u(u_l, u_j)$ are less than the threshold $\delta_u$.

Theorem 3: Our scheme is resistant to SLSA.

Proof: For $u_j$ and $s_j$, every user in $U'$ and their semantics in $S$ meets one of the above two conditions. When we run USR-SLSA algorithm to select users to form the anonymous group, these users whose $d_s(s_i, s_j) \leq \delta_s$ and $d_u(u_l, u_j) \leq \delta_u$ are selected in priority. Then, if less than $k$ users are selected, USR-SLSA algorithm continues to select users whose $d_s(s_i, s_j) \leq \delta_s$ or $d_u(u_l, u_j) \leq \delta_u$ to form the anonymous group until it can select $k$ users. It ensures that the adversary can't infer which is the semantics of location most likely visited by $u_j$. Thus, our scheme can resist SLSA.

## Evaluation setup

Generally, privacy and utility [31, 32] are two significant metrics to measure privacy pretection technology. In this section, we implement experiments on a real-world dataset to evaluate the performance of our scheme in terms of the privacy and the utility.

**Table 2. 15 categories of semantics in Combination Dataset.**

| Category | Name | Category | Distorted Name | Category | Name |
|---|---|---|---|---|---|
| 1 | Public Services | 6 | Professional Services | 11 | Active Life |
| 2 | Information Services | 7 | Financial Services | 12 | Beauty & Spas |
| 3 | Home Services | 8 | Restaurants | 13 | Automotive |
| 4 | Arts & Entertainment | 9 | Shopping | 14 | Health & Medical |
| 5 | Life Services | 10 | Nightlife | 15 | Hotels & Travel |

## Dataset

The Dataset we use for the evaluation is Yelp dataset [33] and is collected from Yelp, which is the largest review site in the United States. It contains 3 types of information: businesses, reviews and user profiles and has been used for many academic researches, such as recommendation system [34], privacy protection [2], sentiment analysis and opinion mining [35]. By pre-processing Yelp dataset, we get a new dataset (called Combination Dataset) containing 264562 valid reviews in 510 cities for evaluating our experiments. We also add a semantic field into the Combination Dataset. In general, the semantic is used to describe the functionality of the business. For example, the semantic 'restaurant' indicates that the POI is a location providing food. In this sense, if a user visits a business, we can describe the user's activity as the semantic of the business. Thus, the semantic in this paper refers to the user's activity. Based on [33], we classify the semantics into 15 categories, as shown in Table 2.

In Combination Dataset, the number of reviews varies significantly in different cities. For example, the city with the least number of reviews only has one review, the city with the most number of reviews has thousands of reviews. It means that our evaluation is easily affected by the extreme reviews in such cities. Hence, we use the median value of the number of reviews in all cities to reduce the impact of such extreme reviews for our evaluation. The median value is a concept in statistics and probability theory. In this paper, the median value refers to the "middle" number, when the number of reviews in all cities are listed in order from smallest to greatest. But, no city has the same number of reviews as the median value. So, we use the data from Las Vegas which has the closest number of reviews to the median. The statistic for Las Vegas is shown in Table 3.

## Experimental settings

In reality, humans are accustomed to periodically engaging in the same activities in the same areas. For example, humans eat lunch near their workplace every weekday. Thus, we partition Las Vegas into 5*5 grids and each grid represents a region. Considering that humans schedule activities based on weekdays and weekends, we set the user's activity cycle based on the week. Assume Alice engages in many activities (visit the businesses) in a grid and the activity 'Shopping' on the 3rd day (Tuesday) of the week appears most frequently. Then we set Alice to engage in the activity 'Shopping' on the Tuesdays of each week.

Intuitively, people are accustomed to engaging in different activities at different times of the day. It is customary to divide time period according to morning—afternoon—evening.

**Table 3. The statistic for Las Vegas in Combination Dataset.**

| Businesses | Users | Reviews | Trajectories |
|---|---|---|---|
| 517 | 81 | 994 | 81 |

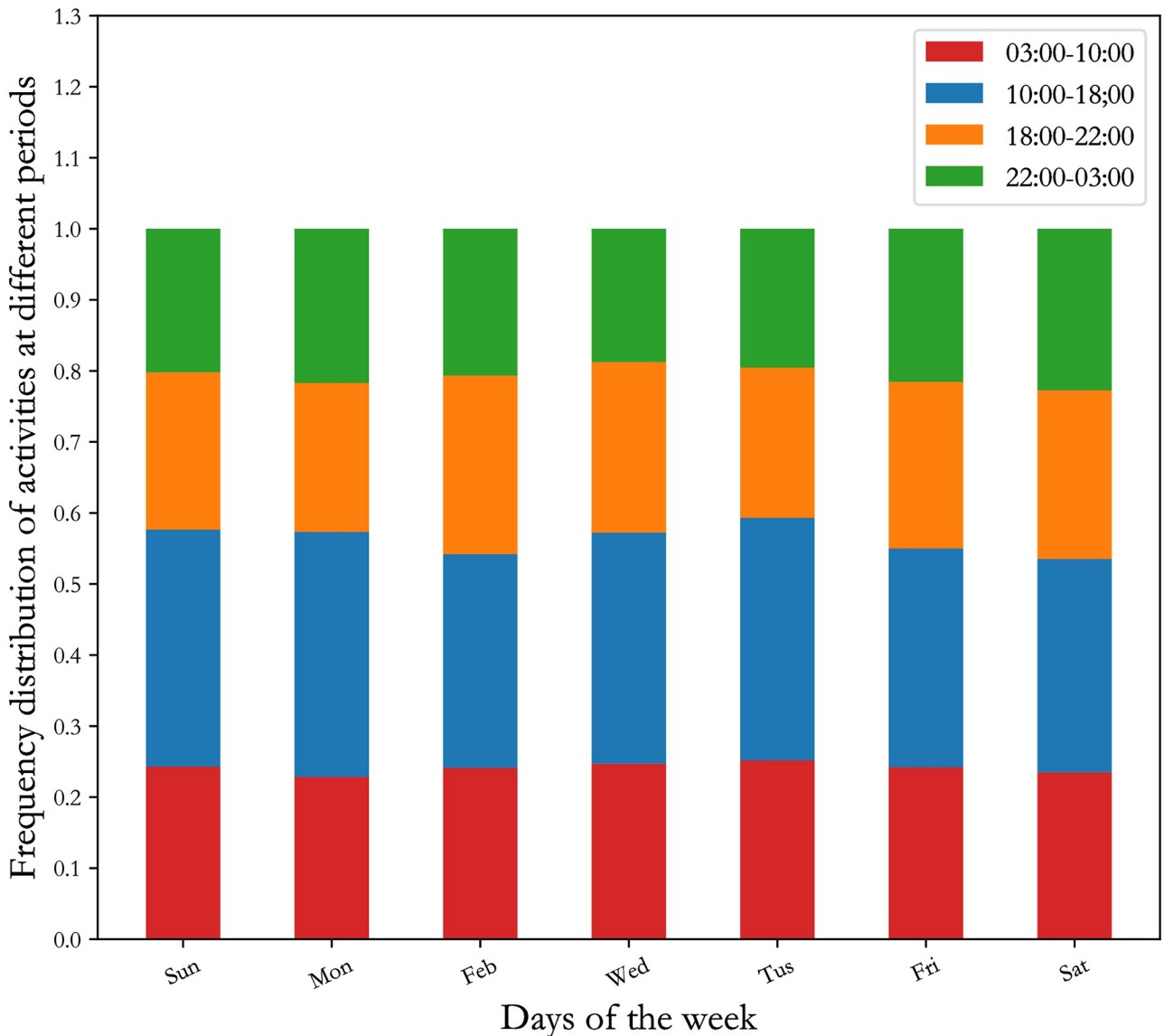

**Fig 4. The frequency distribution of users' activities at different time periods in days of the week.**

Based on this division, we assume that people work in the workplace in the morning and go to nightclubs in the evening. Therefore, the division of time periods will influence the adversary's inference about user's activities. This division is also based on the fact that the frequency of people's activities is stable at different times of the day. To set the time period, we analyze the frequency distribution of users' activities at different time periods on days of the week. Fig 4 shows that the frequency distribution of activities at 4 different periods (03:00-10:00, 10:00-18:00, 18:00-22:00, 22:00-03:00) are stable. On each day of the week, the frequency of users' activities at time periods 03:00-10:00 and 10:00-18:00 steadily falls in the range [0,0.24] and [0.24,0.58], respectively. The frequency of users' activities at time periods 18:00-22:00 and 22:00-03:00 steadily falls in the range [0.58,0.8] and [0.8,1], respectively. Thus, we divide the cycle into 4 time periods: 03:00-10:00, 10:00-18:00, 18:00-22:00 and 22:00-03:00.

## Evaluation metric

**Privacy metric.** The goal of the adversary is to get users' original trajectories. To do this, the adversary will exploit a particular location or series of locations he has known exclusively for a user to reconstruct the original trajectories. The more locations the adversary knows, the more likely he is to reconstruct them. Hence, the adversary will use different ways to get as many exclusive locations of a user as possible. For example, launching SLSA to find that a location with the most frequently occurring semantic is the user's real location. Based on Section Motivation and Basic idea, the *distortion* metric can be used to quantify the privacy. For a user and one of her trajectories, if its *distortion* is greater than $\delta_j$, the adversary can reconstruct it, i.e., the privacy is compromised.

For our scheme, among all original trajectories, the more trajectories with the *distortion* not greater than $\delta_j$, the better the privacy protection of this scheme. Thus, we use the ratio of original trajectories whose corresponding *distortion* is not greater than $\delta_j$ to quantify the privacy-preserving efficiency of our scheme (called *effective distortion ratio*).

It includes 4 cases in which the adversary can reconstruct the user's original trajectory: 1. Users do not exchange reviews with other users, which allows the adversary to directly obtain the original trajectory; 2. reviews are exchanged between users, but RLCA is not considered; 3. RLCA is considered, but SLSA is not; 4. both RLCA and SLSA are considered. To evaluate the impact of these different cases on user privacy, we compare USR-SLSA algorithm with RUS, USR-RLCA, the non-exchange review solution (Non-exchange) and the theoretically optimal solution (Optimal). Non-exchange corresponds to the case1. Optimal will lead to a theoretically optimal result that the adversary can't infer any original trajectories.

**Utility metric.** Because users submit reviews to SP primarily for publishing, we must consider the user utility of users in terms of publication. Paper [2] is the first study on review publishing considering system utility, personal profile, and privacy in multiple regions and can preserve user location privacy by suppressing some public reviews. Since our scheme does not focus on how to publish reviews, we use $(\epsilon, \delta)$-public principle, which is a review publication mechanism used in the literature [2], to publish reviews. $\epsilon$ and $\delta$ are thresholds and are used to balance the number of anonymous reviews and the number of public reviews for each business. In the mechanism, all reviews are public when the number of the reviews for $L_i$ ($L_i$ refers to a business.) is less than $\delta$. At least $\epsilon$ out of top-$\delta$ useful reviews are public when the number of the reviews for $L_i$ is no less than $\delta$. The mechanism can preserve users' location privacy by suppressing some public reviews.

As mentioned in the Introduction, users hope to build reputations for POIs by publishing reviews. To ensure a more objective reputation, users want to publish as many reviews as possible. Therefore, we define the user utility as users' reviews that are published and measure the utility as the ratio of the public reviews. Public review is a metric used in the literature [2] and refers to the number of all users' published reviews. Public reviews increase as the global budget increases. Global budget refers to the maximum number of reviews that can be published by every user in all regions. To evaluate the user utility, we compare our scheme with the method of literature [2] (LRPM) and we set $\epsilon = 2, \delta = 3$ and the global budget ranges from 20 to 70.

## Results

1. Privacy-preserving efficiency. We first evaluate the privacy-preserving efficiency of the USR-SLSA for our scheme. Due to the impact of different parameters on the privacy-preserving efficiency, we separately evaluate the impact of $k$, $\delta_j$, $\delta_u$, and $\delta_s$ on the privacy-

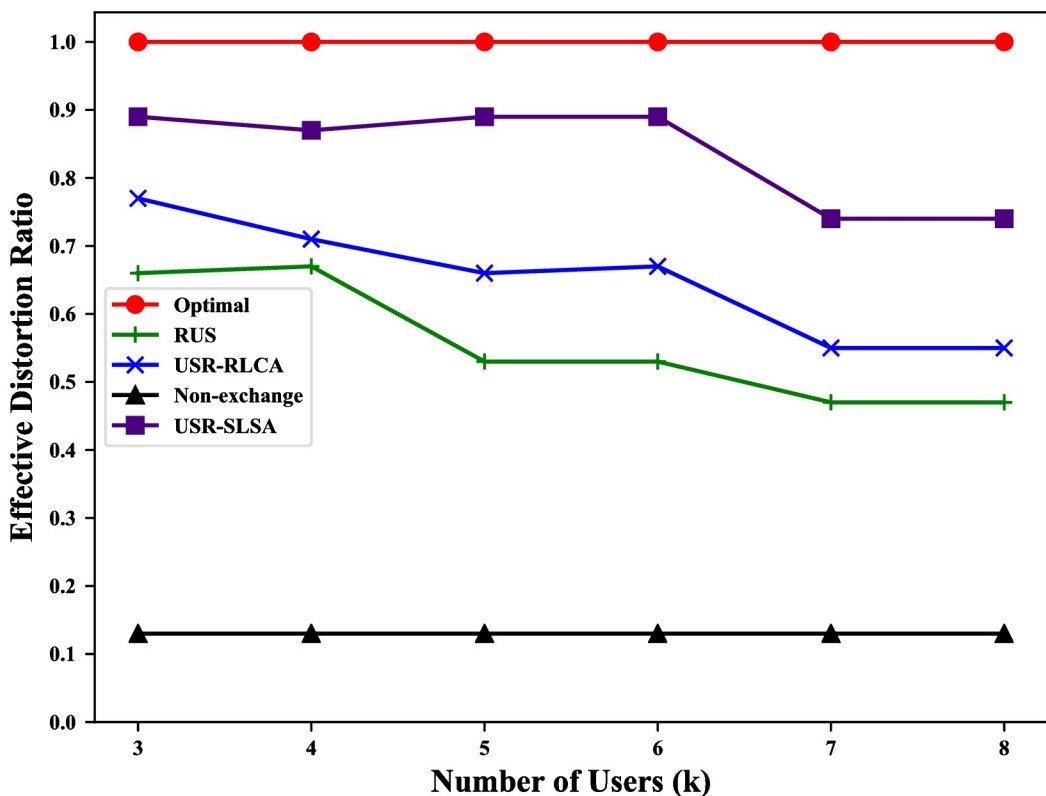

**Fig 5. Impact of *k* on the privacy-preserving efficiency.**

preserving efficiency. Figs 5–8 show the *effective distortion ratio* for five compared algorithms with different parameters, respectively.

Specifically, Fig 5 shows the change of *effective distortion ratio* when *k* increases. Note that only reviews with the same POI, time period, etc., can be exchanged. Therefore, in our dataset, only no more than 8 users can form an anonymous group. We can see that the *effective distortion ratio* for RUS, USR-RLCA and USR-SLSA slowly reduces with the growth of *k* when $k < 7$ and remains constant when $k \geq 7$. This is because the number of anonymous groups that can contain at least *k* reviews reduces as *k* increases.

Fig 6 shows the change of *effective distortion ratio* when $\delta_j$ increases. As shown in Fig 6, the *effective distortion ratio* for three algorithms hardly changes with the increase of $\delta_j$ when $\delta_j > 0.7$ (the corresponding $\delta_j$ for RUS, USR-RLCA and USR-SLSA range from 0.7 to 1.0, 0.8 to 0.9 and 0.8 to 1.0, separately). $\delta_j$ can determine the privacy-preserving efficiency for three different algorithms only when $\delta_j$ is less than 0.7.

From Figs 7 and 8, we observe that the *effective distortion ratio* of three algorithms hardly changes when $\delta_s(\delta_u)$ is more than 0.7. When we set $\delta_u = 0.5$, their *effective distortion ratio* increases as $\delta_s$ grows when $\delta_s$ is more than 0.5. When we set $\delta_s = 0.5$, their *effective distortion ratio* increases as $\delta_u$ grows when $\delta_u$ is more than 0.5. This is because some users only submit 1 or 2 reviews. Considering the occasionality and randomness of user behavior, adversaries cannot exploit such reviews to obtain the privacy of the corresponding users, which lead to an increase in the *effective distortion ratio*.

Figs 5–8 also shows some similar evaluation results. Firstly we can see that the *effective distortion ratio* of USR-RLCA is larger than that of RUS. The reason is that, compared with

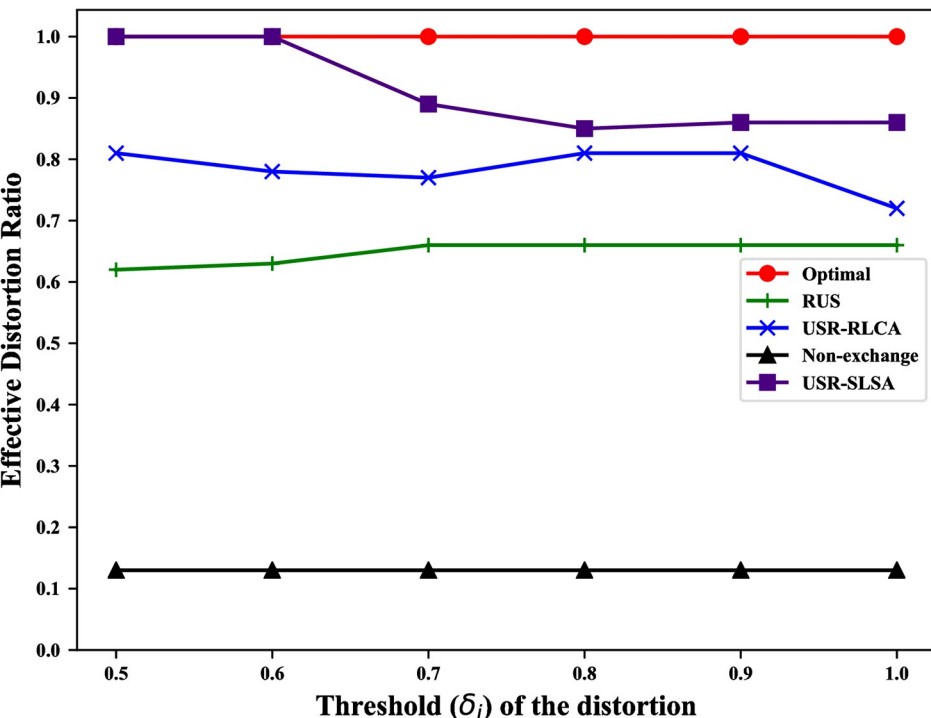

**Fig 6. Impact of $\delta_j$ on the privacy-preserving efficiency.**

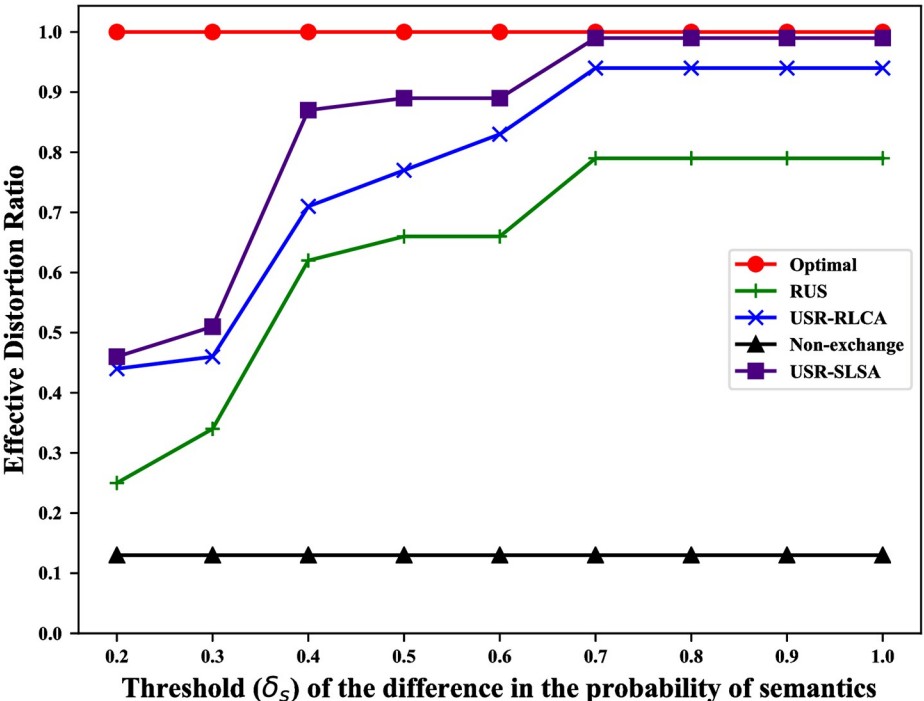

**Fig 7. Impact of $\delta_s$ on the privacy-preserving efficiency.**

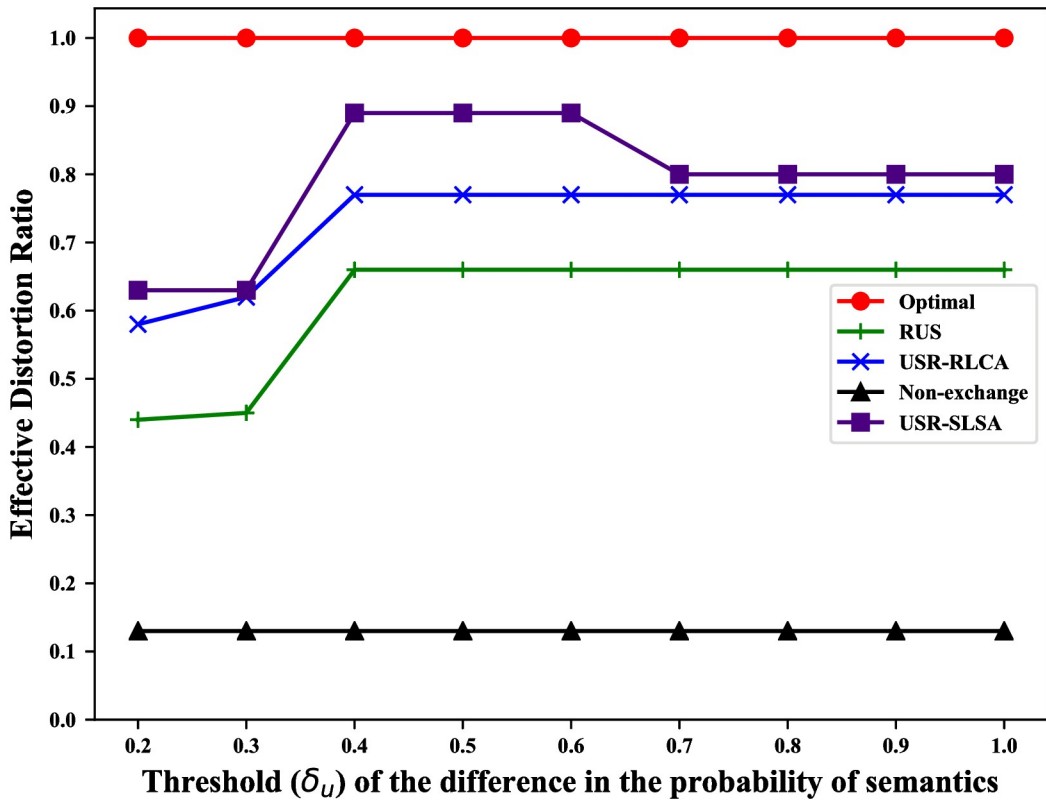

**Fig 8. Impact of $\delta_u$ on the privacy-preserving efficiency.**

RUS, the adversaries cannot identify more trajectories by launching RLCA when the reviews are exchanged using USR-RLCA. It proves that USR-RLCA can resist RLCA. The *effective distortion ratio* of USR-SLSA is larger than that of USR-RLCA. The reason is that USR-SLSA is resistant to only both RLCA and SLSA. As a result, USR-SLSA enables fewer trajectories to be identified by the adversafy than USR-RLCA. Secondly, the result of Non-exchange shows that even if users do not exchange reviews, adversaries cannot gain all users' privacy. As in the above analysis, adversaries cannot infer the privacy of users who submit only 1 or 2 reviews. However, such reviews can be exploited to some extent by adversaries to identify the trajectories of other users. As a consequence, the *effective distortion ratio* of USR-SLSA is always less than 1.0. Thirdly, since some reviews will be exchanged by performing RUS, the *effective distortion ratio* of Non-exchange is lower than that of RUS. Besides, we also observe that the *effective distortion ratio* of USR-SLSA is larger than that of USR-RLCA. This is because USR-SLSA can resist RLCA and SLSA while USR-RLCA can only resist RLCA.

2. User utility. In this part, we evaluate user utility in the case where the SPs receive reviews sent by USR-SLSA and publish them by the $(\epsilon, \delta)$-public principle. Fig 9 shows the ratio of public reviews for USR-SLSA and LRPM for different global budget. We observe that USR-SLSA has a larger ratio of public reviews than LRPM. Because the SPs receive fewer reviews published by USR-SLSA than LRPM. Thus, for the same global budget, USR-SLSA can publish a larger percentage of public reviews. However, Fig 9 does not sufficiently illustrate that USR-SLSA can publish a larger number of reviews than LRPM. Therefore, we further evaluate the ratio of the number of reviews published by USR-SLSA to all reviews. The

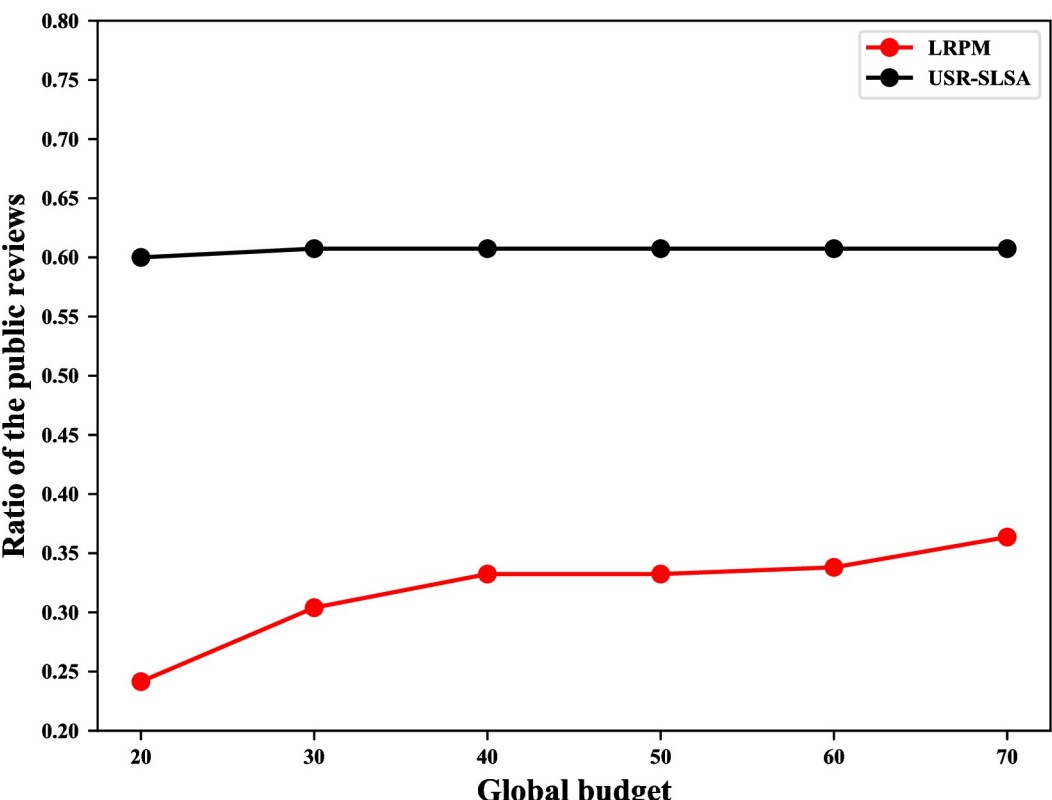

**Fig 9. Ratio of the public reviews for different global budget (SPs for two methods receive different numbers of reviews).**

evaluation results are shown as Fig 10. The ratio of public reviews is almost identical for both of them. Because the reviews submitted by the users to the SPs through USR-SLSA cannot reveal the privacy, thus the users can publish more reviews.

## Conclusion

In this paper, we study the exchanging reviews for trajectory privacy protection in LBSSs. Since the LBSS is a registration system, adversaries can easily obtain user profiles and trajectories embedded in the reviews submitted to the SPs by compromising with the SPs. To protect trajectory privacy, we propose an approach to exchanging reviews before users submitting reviews to the SPs. However, after analysis, we find that exchanging reviews can be easily broken by RLCA and SLSA if we randomly exchange users' reviews. To resist the two attacks, we design two schemes named USR-RLCA and USR-SLSA to exchange reviews. For USR-RLCA, we propose a metric to measure the correlation between a user and a trajectory. Based on the metric, USR-RLCA can select reviews resisting RLCA to exchange by suppressing the number of locations on each reconstructed trajectory below a threshold. For USR-SLSA, we propose a metric to measure the indistinguishability of locations concerning the difference of semantic frequency in a long term. Based on the metric, USR-SLSA can select reviews resisting RLCA to exchange by allowing two reviews, which the probability difference of their semantics is below a threshold after the exchange, to be exchanged. The evaluation results demonstrate that our approach can effectively protect trajectory privacy when real-name users submit their reviews to SPs and do not degrade users' utility in terms of review publication.

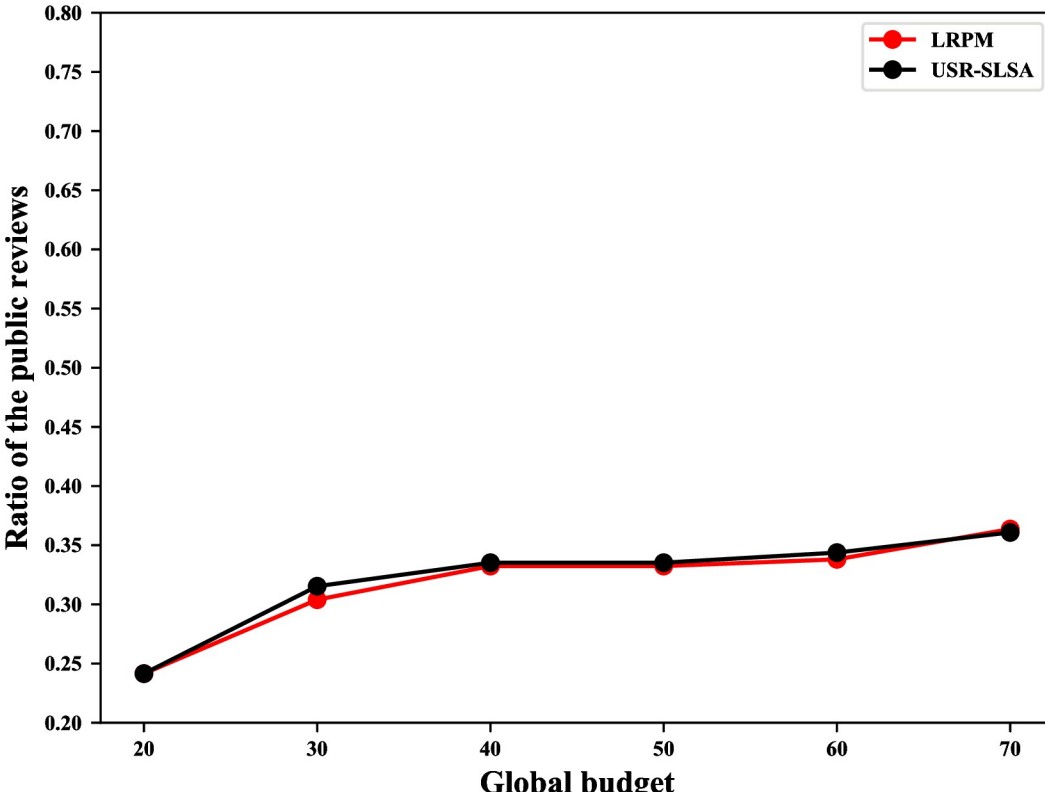

**Fig 10. Ratio of the public reviews for different global budget (SPs for two methods receive the same numbers of reviews).**

Yet in fact, our study is based on two assumptions: (1) users are registered in real names on LBSSs; (2) a user is allowed to review on businesses he has enjoyed the service. There are still some LBSSs that do not require users to register with real names or allow users to review on businesses without restrictions. This enhances the complexity of the exchange of reviews and privacy protection would be a more interesting and challenging topic. Our future will focus on how to exchange reviews in such scenarios.

## Supporting information

**S1 Dataset.**
(CSV)

## Author Contributions

**Conceptualization:** Yunfeng Wang.

**Data curation:** Yunfeng Wang, Mingzhen Li, Qifeng Tang.

**Funding acquisition:** Yang Xin, Yixian Yang, Yuling Chen.

**Investigation:** Yunfeng Wang.

**Methodology:** Yunfeng Wang, Mingzhen Li.

**Supervision:** Yang Xin, Hongliang Zhu, Yixian Yang, Yuling Chen.

**Validation:** Yang Xin, Qifeng Tang, Hongliang Zhu.

**Writing – original draft:** Yunfeng Wang.

**Writing – review & editing:** Mingzhen Li, Yang Xin, Guangcan Yang, Hongliang Zhu, Yixian Yang, Yuling Chen.

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
