## [Decision Letter · Decision Letter 0]

24 Jun 2021

PONE-D-21-10068

Exchanging registered users'  submitting reviews towards trajectory privacy preservation for review services in Location-Based Social Networks

PLOS ONE

Dear Dr. Wang,

Thank you for submitting your manuscript to PLOS ONE. After careful consideration, we feel that it has merit but does not fully meet PLOS ONE’s publication criteria as it currently stands. Therefore, we invite you to submit a revised version of the manuscript that addresses the points raised during the review process.

We look forward to receiving your revised manuscript.

Kind regards,

Hua Wang

Academic Editor

PLOS ONE

“This work was partially supported by the National Key R&D Program of China under 911 Grant 2017YFB0802300, in part Major Scientific and Technological Special Project of 912 Guizhou Province under Grant 20183001,Foundation of Guizhou Provincial Key 913 Laboratory of Public Big Data under Grant 2018BDKFJJ008 and Grant 914 2018BDKFJJ021, National Statistical Scientific Research Project of China under Grant 915 2018LY61 and Grant 2019LY82, and in part Research Project of Hechi University under 916 Grant 2021KY0615 and 2021KY0620.”

“This work was partially supported by the National Key R&D Program of China under 911 Grant 2017YFB0802300, in part Major Scientific and Technological Special Project of 912 Guizhou Province under Grant 20183001,Foundation of Guizhou Provincial Key 913 Laboratory of Public Big Data under Grant 2018BDKFJJ008 and Grant 914 2018BDKFJJ021, National Statistical Scientific Research Project of China under Grant 915 2018LY61 and Grant 2019LY82, and in part Research Project of Hechi University under 916 Grant 2021KY0615 and 2021KY0620.”

Additional Editor Comments (if provided):

Please address all questions and concerns raised by the reviewers.

Reviewers' comments:

Reviewer's Responses to Questions

**Comments to the Author**

1. Is the manuscript technically sound, and do the data support the conclusions?

Reviewer #1: Yes

Reviewer #2: Yes

2. Has the statistical analysis been performed appropriately and rigorously? 

Reviewer #1: Yes

Reviewer #2: Yes

3. Have the authors made all data underlying the findings in their manuscript fully available?

Reviewer #1: Yes

Reviewer #2: Yes

4. Is the manuscript presented in an intelligible fashion and written in standard English?

Reviewer #1: Yes

Reviewer #2: No

5. Review Comments to the Author

Reviewer #1: Comments:

The authors proposed a method based on exchangingreviews to achieve trajectory privacypreservation. Users’ reviews are exchanged before they submits them to the service providers.The idea of disrupting the correspondence between users and reviews by exchanging reviews is good. It makes it impossible for adversaries to obtain trajectory privacy even if they get users’ profiles and reviews. The presentation is good. However, there exit some minor problems.

(1) Some sentences contain minor grammatical errors and need to be checked.

(2) Page 25, ‘’Fig 5,Fig 6,Fig 7,Fig 8’’, add“ and ” to separate Fig 7 and Fig 8.

(3) Table2 is incomplete, Please redraw it.

(4) Check the format carefully throughout the paper according to the template.

Reviewer #2: In this paper, authors study the exchange of reviews for trajectory privacy protection in LBSNs.

The topic of this paper is interesting. There may have some concerns to address in the future revision:

(1) Specific technology or method of the designed two schemes may be include in abstract.

(2) What the meaning of the sentence Line63-43 Page3? It seems to be a contradiction technically feasible and revealing users' privacy.

(3) In Definition 2, the author defines a three-tuple<poi(ri,j), t="">. However, I never found the use of variable t(ri,j) again. I guess it may be used sort the set of locations in algorithms. If it’s true, please adding more description in algorithms.

(4) What the meaning of median in Page22 Line766? Why you only choose the data from Las Vegs?

(5) In subsection Utility metric, the author points out that (ϵ,δ) is a review publication mechanism used in the literature [2] and set the value of the experiment. But I cannot find out what each of notation means.

(6) In Fig5, k respects number of users, the manuscript emphasizes more users in Location-Based Social Networks, but only 3 to 8 people were involved in your experiment.

(7) It is noted that this paper needs more careful editing, such as grammar, and sentence structure, etc., for example:

Line285 Page8, TJ should be Tj.

Line360 Page9, ∑_(i=2)^m s_i×n_i should be ∑_(i=2)^m n_i .

Line362 Page9, s1 should be si.

Line801 Page24, the article has no “Section 3.3”, authors should notice the form of expression.

Line844-845 Page24, this sentence lacks the conjunction.

Line847 Page25, e effectivedistortionratio should be modified.

Line891 Page26, “In” should be “in”.

Table 2 is incomplete.</poi(ri,j),>

6. PLOS authors have the option to publish the peer review history of their article (what does this mean?). If published, this will include your full peer review and any attached files.

Reviewer #1: No

Reviewer #2: No

---

## [Author Response · Author response to Decision Letter 0]

23 Jul 2021

Editor, Concern # 1: Please ensure that your manuscript meets PLOS ONE's style requirements, including those for file naming. The PLOS ONE style templates can be found at.

Author response: Thank you very sincerely for giving us your comments to improve our paper.

Our manuscript has been edited in accordance with the format of PLOS ONE's and we sure that our manuscript meets PLOS ONE's style requirements, including those for file naming.

Editor, Concern #2: We note that the grant information you provided in the ‘Funding Information’ and ‘Financial Disclosure’ sections do not match. When you resubmit, please ensure that you provide the correct grant numbers for the awards you received for your study in the ‘Funding Information’ section.

Author response: Thank you very much for your comment. We have removed the grant information in the Acknowledgments Section of our manuscript. We also re-edited our grant information and added a ‘funding information updated statement’ to the Cover letter.

Editor, Concern # 3: Thank you for stating the following in the Acknowledgments Section of your manuscript:“This work was partially supported by the National Key R&D Program of China under 911 Grant 2017YFB0802300, in part Major Scientific and Technological Special Project of 912 Guizhou Province under Grant 20183001,Foundation of Guizhou Provincial Key 913 Laboratory of Public Big Data under Grant 2018BDKFJJ008 and Grant 914 2018BDKFJJ021, National Statistical Scientific Research Project of China under Grant 915 2018LY61 and Grant 2019LY82, and in part Research Project of Hechi University under 916 Grant 2021KY0615 and 2021KY0620.”

Author response: Thank you very much for your comment. We have removed the grant information in the Acknowledgments Section of our manuscript. We also re-edited our grant information and added a ‘funding information updated statement’ to the Cover letter. The updated funding information is as follows:

This research was funded by the “Major Scientific and Technological Special Project of Guizhou Province (20183001)”, the “Foundation of Guizhou Provincial Key Laboratory of Public Big Data (2017BDKFJJ015, 2018BDKFJJ008, 2018BDKFJJ020, 2018BDKFJJ021)”, and the “Basic Ability Improvement Program for Young and Middle-aged Teachers of Guangxi(2021KY0615 and 2021KY0620)”.

Editor, Concern # 4: Please review your reference list to ensure that it is complete and correct. If you have cited papers that have been retracted, please include the rationale for doing so in the manuscript text, or remove these references and replace them with relevant current references. Any changes to the reference list should be mentioned in the rebuttal letter that accompanies your revised manuscript. If you need to cite a retracted article, indicate the article’s retracted status in the References list and also include a citation and full reference for the retraction notice.

Author response: Thank you very much for your comment. According to the comment, we consulted more literatures and further improved the paper.

Reviewer#1, Concern # 1: Some sentences contain minor grammatical errors and need to be checked.

Author response: Thank you very much for your comment. Before we answer the following questions, we first thank you very sincerely for agreeing with my idea of our paper. According with your advice, we checked our paper carefully and modified the mistakes of grammars sentence by sentence.

Reviewer#1, Concern # 2: Page 25, ‘’Fig 5,Fig 6,Fig 7,Fig 8’’, add“ and ” to separate Fig 7 and Fig 8.

Author response: Thank you very much for your comment. We checked our paper carefully and modified the mistake on Page 24 and 25.

Author action: We updated the manuscript by adding the word ‘and’ between ‘Fig 7’ and ‘Fig 8’. 

Reviewer#1, Concern # 3: Table2 is incomplete, Please redraw it.

Author response: Thank you very much for your comment. We checked Table 2 in our paper carefully and redraw it.

Author action: We updated the manuscript by redrawing Table2. In the original manuscript, Table 2 includes 6 columns, but only 5 columns are displayed. Through modification, all columns are displayed in the revised manuscript.

Reviewer#1, Concern # 4: Check the format carefully throughout the paper according to the template.

Author response: Thank you very much for your comment. We checked the format carefully according to the template and made some modifications which are inconsistent with the format of the template. For example, delete redundant paragraphs that is not related to the manuscript, modify titles of subsections, delete the section Acknowledgment, redraw Table 2, and reedit the format of some references.

Author action: We updated the manuscript by deleting redundant paragraphs, modifying titles of subsections and deleting the section Acknowledgment, redraw Table 2, and reedit the format of some references.

Reviewer#2, Concern # 1: Specific technology or method of the designed two schemes may be include in abstract.

Author response: We sincerely thank you for giving us the comment. The comment is valuable for expressing our schemes clearly. We reorganized and rewrote the sentences in Section Abstract. We also rewrote some sentences related to the comment in Section Introduction and Section Conclusion.

Author action: We updated the manuscript by reorganizing and rewriting the description related to the comment in 3 sections: Abstract, Introduction and Conclusion.

Reviewer#2, Concern # 2: What the meaning of the sentence Line63-43 Page3? It seems to be a contradiction technically feasible and revealing users' privacy.

Author response: Thank you very much for your comment. I am sorry for not clearly and accurately stating the meaning of the sentence Line63-43 Page3 in the original manuscript, due to our grammar and language issues. We rewrote these sentences and ensured that they could clearly and accurately state their meaning.

Author action: We updated the manuscript by rewriting these sentences Line63-43 Page3 in the original manuscript and ensured that these sentences could clearly and accurately state their meaning. 

Reviewer#2, Concern # 3: In Definition 2, the author defines a three-tuple. However, I never found the use of variable t(r_ij) again. I guess it may be used sort the set of locations in algorithms. If it’s true, please adding more description in algorithms.

Author response: Thank you very much for your comment. The use of variable t(r_ij) involves two contents in the manuscript. One is used to define the trajectory T_j in Definition 2. In this definition, a trajectory T_j of u_j is a sequence of locations sorted in chronological order in which u_j has visited and reviewed the POI(r_ij). Another is used to sort the set of locations in algorithms. So, we added the description to state the use of variable t(r_ij) in Section-The algorithm design and in the Section-System model and basic concepts.

Author action: We updated the manuscript by adding the more description to state the use of variable t(r_ij). In Section-The algorithm design, we added the description to state how the set of locations are be sorted using variable t(r_ij). In Section- System model and basic concepts, we also added the description to clarify the use of variable t(r_ij) in Definition 2, namely a trajectory is a sequence of locations sorted in chronological order in which they are reviewed.

Reviewer#2, Concern # 4: What the meaning of median in Page22 Line766? Why you only choose the data from Las Vegs?

Author response: Thank you very much for your comment. The comment pointed out the problem that we did not clearly state the reason why we only choose the data from Las Vegs. It can help to improve our paper by clearly stating the reason. In our manuscript, “median” refers to the median value. The median value is a concept in statistics and probability theory. For a finite list of numbers, the median value is the "middle" number, when those numbers are listed in order from smallest to greatest. The advantage of the median value is that it is not skewed by a small proportion of extremely large or small values, and therefore provides a better representation of most of the numbers in the finite list.

Author action: We updated the manuscript by adding the description to state what the meaning of median is and the reason why we only choose the data from Las Vegs.

Reviewer#2, Concern # 5: In subsection Utility metric, the author points out that (ϵ,δ) is a review publication mechanism used in the literature [2] and set the value of the experiment. But I cannot find out what each of notation means. 

Author response: Thank you very much for your comment. ϵ and δ are notations in literature [2]. They are thresholds and are used to balance the number of anonymous reviews and the number of public reviews for each business. In the revised manuscript, we added the description in Section- Utility metric to clarify: (1) the means of ϵ and δ are; (2) how ϵ and δ are used to preserve user location privacy; (3) the reason why we use them to publish reviews.

Author action: We updated the manuscript by adding the description in Section- Utility metric to state (1) what ϵ and δ means; (2) how ϵ and δ are used to preserve user location privacy; (3) why we use (ϵ,δ)-public principle to publish reviews.

Reviewer#2, Concern # 6: In Fig5, k respects number of users, the manuscript emphasizes more users in Location-Based Social Networks, but only 3 to 8 people were involved in your experiment.

Author response: Thank you very much for your comment. I am sorry for not clearly stating the meaning of the notation k. k only represents the number of users included in an anonymous group, not the number of all users in the social network. In this response, we attempt to give an example to illustrate k. Assume that there are 10 users in a social network and each user has a trajectory with 10 locations. All locations in these trajectories are different. For RLCA, we assume δ_j=0.5. For user u_j, δ_j=0.5 means that the adversary can obtain the trajectory T_u by launching the RLCA if T_u and its distorted trajectory T_u^' have more than 5 locations in common after exchanging reviews. We consider that u_j only exchanges review with one user. Then we find that T_u and T_u^' always have at least 5 locations in common after exchanging reviews. So u_j should exchange reviews with as many users as possible. However, more users always lead to higher overload due to the selection of more users. Therefore, we set the security parameter k to represent the number of users which exchange reviews with u_j. These users form an anonymous group. In addition, in our dataset, affected by the number of reviews of different users, the time, period, cycle, only 3-8 users can form an anonymous group. So, we added the description in Section- Results to clarify the reason why only 3 to 8 people were involved in our experiment

Author action: We updated the manuscript by adding the description in Section- Results clarify the reason why only 3 to 8 people were involved in our experiment.

Reviewer#2, Concern # 7: It is noted that this paper needs more careful editing, such as grammar, and sentence structure, etc., for example: (1) Line285 Page8, T_J should be T_j; (2) Line360 Page9, ∑_(i=2)^m▒〖s_i×n〗_i should be ∑_(i=2)^m▒n_i ; (3) Line362 Page9, s_1 should be s_i; (4) Line801 Page24, the article has no “Section 3.3”, authors should notice the form of expression; (5) Line844-845 Page24, this sentence lacks the conjunction; (6) Line847 Page25, e effective e distortion ratio should be modified; (7) Line891 Page26, “In” should be “in”; (8) Table 2 is incomplete.

Author response: Thank you very much for your comment. The comment is very helpful for revising and improving our paper. We carefully checked grammar, sentence structure, etc., throughout the paper. The specific modification is as follows: (1) We made some modifications which are inconsistent with the format of the template; (2) We modified the mistakes of grammars and sentence structure sentence by sentence.

Author action: We updated the manuscript by modifying the mistakes of the format, grammars and sentence structure.

---

## [Decision Letter · Decision Letter 1]

18 Aug 2021

Exchanging registered users'  submitting reviews towards trajectory privacy preservation for review services in Location-Based Social Networks

PONE-D-21-10068R1

Dear Dr. Wang,

We’re pleased to inform you that your manuscript has been judged scientifically suitable for publication and will be formally accepted for publication once it meets all outstanding technical requirements.

Kind regards,

Hua Wang

Academic Editor

PLOS ONE

Additional Editor Comments (optional):

Reviewers' comments:

Reviewer's Responses to Questions

**Comments to the Author**

1. If the authors have adequately addressed your comments raised in a previous round of review and you feel that this manuscript is now acceptable for publication, you may indicate that here to bypass the “Comments to the Author” section, enter your conflict of interest statement in the “Confidential to Editor” section, and submit your "Accept" recommendation.

Reviewer #1: All comments have been addressed

Reviewer #2: All comments have been addressed

2. Is the manuscript technically sound, and do the data support the conclusions?

Reviewer #1: Yes

Reviewer #2: Yes

3. Has the statistical analysis been performed appropriately and rigorously? 

Reviewer #1: Yes

Reviewer #2: Yes

4. Have the authors made all data underlying the findings in their manuscript fully available?

Reviewer #1: Yes

Reviewer #2: Yes

5. Is the manuscript presented in an intelligible fashion and written in standard English?

Reviewer #1: Yes

Reviewer #2: Yes

6. Review Comments to the Author

Reviewer #1: As a revised version I think the questions that I had found are all revised and this manuscript can be accepted.

Reviewer #2: (No Response)

7. PLOS authors have the option to publish the peer review history of their article (what does this mean?). If published, this will include your full peer review and any attached files.

Reviewer #1: No

Reviewer #2: No

---

## [Editor Report · Acceptance letter]

3 Sep 2021

PONE-D-21-10068R1 

Exchanging registered users' submitting reviews towards trajectory privacy preservation for review services in Location-Based Social Networks 

Dear Dr. Wang:

I'm pleased to inform you that your manuscript has been deemed suitable for publication in PLOS ONE. Congratulations! Your manuscript is now with our production department. 

Kind regards, 

on behalf of

Dr. Hua Wang 

Academic Editor

PLOS ONE